# Use of delayed ERA5-Land soil moisture products for improving landslide early warning

Nunziarita Palazzolo[1], Antonino Cancelliere[1], Robert D. Zofei[1], David J. Peres[1]

[1]Department of Civil Engineering and Architecture, University of Catania, Catania, 95125, Italy

5 *Correspondence to*: Nunziarita Palazzolo (nunziarita.palazzolo@unict.it)

**Abstract.** Previous studies have demonstrated that incorporating soil moisture information in landslide-triggering thresholds can improve their predictive performance. ECMWF ERA5-Land reanalysis soil moisture information has proven to have this potential. However, ERA5-Land data are released with a latency of usually five days, which limits their immediate operational use in Landslide Early Warning Systems (LEWSs). In this study, we investigate whether delayed soil moisture data—ranging from 0 to 15 days prior to rainfall events—can still effectively inform landslide-triggering conditions. Specifically, we develop artificial neural networks (ANNs) trained on various delay times and evaluate how detection performances vary with increasing lag. We measure performances by ROC-based indices, such as the True Skill Statistic (TSS). Focusing on Sicily, Italy, our results show that even delayed soil moisture data consistently outperform models based solely on rainfall (TSS = 0.68 vs. 0.59). Notably, TSS reduces only marginally, from 0.78 with no delay to 0.72 with five-day delay, and 0.67 with fifteen-day delay. This performance remains higher than that obtained using only soil moisture data (without precipitation and no delay, TSS = 0.53), as well as those achieved with a traditional power-law threshold based on rainfall intensity and duration (TSS = 0.50) and also through ANN model using rainfall intensity and duration (TSS = 0.59). These findings are, thus, promising for an operational use of ERA5-Land soil moisture products in LEWSs.

## 1 Introduction

20 Rainfall-induced landslides rank among the most devastating natural hazards, causing extensive loss of life and severe impacts on the environment, infrastructure, and the economy (Winter et al., 2016). Between 2004 and 2016, 4,862 fatal landslides were recorded worldwide, leading to an estimated 55,997 fatalities (Froude and Petley, 2018). Territorial Landslide Early Warning Systems (LEWSs) aim to promptly alert at-risk populations, thereby significantly reducing casualties and injuries. Identifying the conditions that trigger landslides is crucial for developing effective LEWSs. Early 25 efforts primarily focused on rainfall intensity–duration thresholds (Caine, 1980; Guzzetti et al., 2007), but more recent studies highlight the value of incorporating additional hydrological variables—particularly soil moisture—to better capture the mechanisms driving slope failure (Uwihirwe et al., 2022; Mirus et al., 2018a, b; Thomas et al., 2018; Segoni et al., 2018; Wicki et al., 2021; Bogaard and Greco, 2016, 2018; Reder and Rianna, 2021; Marino et al., 2020; Conrad et al., 2021; Palazzolo et al., 2023; Meneses et al., 2019).

Machine learning methods, particularly Artificial Neural Networks (ANNs), have proven effective in handling multiple predictors for landslide forecasting, often going beyond classical parametric threshold models (e.g., power-law relationships).

Distefano et al., (2022), focusing on Sicily, showed that precipitation thresholds derived via Artificial Neural Networks (ANNs) outperform traditional power-law formulations. This evidence was further supported at broader scales. Mondini et al., (2023) demonstrated that a simple deep learning framework, based on a feed-forward network with two hidden layers of four neurons and bagging for robustness, could reliably forecast rainfall-induced landslides across Italy, achieving AUC values between 0.88 and 0.92. Alongside, Peng and Wu, (2024) applied multilayer perceptron (MLP) regression to extend classical effective rainfall–duration thresholds by including daily rainfall and showed that the proportion of hazardous conditions classified as warning or severe warning increased markedly, from 41.5% to 76.8%. Expanding the comparison across algorithms, Harsa et al., (2023) tested Random Forest (RF), Extreme Gradient Boosting (XGBoost), Generalized Boosting Machine (GBM), Generalized Linear Model (GLM), and a feed-forward Deep Learning network with satellite-derived rainfall in Central Java (Indonesia) to build landslide event prediction models using as input of the algorithms precipitation data obtained from the global satellite mapping. They found that the GLM performed best on raw data (AUC = 0.828), while the Deep Learning model achieved the best results after log-transformation (AUC = 0.836). More recently, the focus shifted toward both predictive skill and interpretability. Shao et al. (2025) employed XGBoost together with SHAP (SHapley Additive exPlanations) to integrate rainfall, soil moisture, and environmental factors in Italy, reporting excellent performance (AUC = 0.917 ± 0.026, sensitivity = 0.79, specificity = 0.81), and outperforming traditional E–D curves. Similarly, Wen et al., (2025) used CatBoost regression with SHAP in Fengjie County (China), highlighting daily rainfall (DR) and early effective rainfall (ER) as key variables, and identifying critical values around DR ≈ 5 mm and ER ≈ 15 mm above which landslide probability increases significantly ($R^2$ = 0.987; RMSE = 0.075). Finally, Lee et al., (2025) developed a nationwide framework for South Korea using AutoML-optimized RF models with SHAP interpretation, obtaining significant high predictive skill (AUC = 0.981; accuracy = 0.93), and showing that 88% of landslides in 2023 were classified as high or very high susceptibility, supporting the model's use in daily warning up to 72 h in advance.

Together, these studies illustrate a progression from traditional rainfall thresholds toward increasingly complex machine learning frameworks for operational landslide early warning. Meanwhile, other studies have explored the benefit from the incorporation of soil-hydrologic monitoring data into ML-based models to simulate changes in soil moisture and pore pressure that predispose triggering mechanisms. Orland et al., (2020), for instance, developed a deep learning model that combined soil moisture, pore pressure, and rainfall monitoring data from landslide-prone hillslopes in the USA to predict both the timing and magnitude of hydrologic responses at various soil depths. Their findings suggest that machine learning can provide an accurate and computationally efficient alternative to empirical approaches and physical models for landslide hazard warning. Along the same line, Distefano et al. (2023) showed that incorporating ERA5-Land reanalysis soil moisture products (Muñoz-Sabater et al., 2021)—specifically the soil moisture at the onset of rainfall events—can significantly improve ANN-based landslide predictions. However, since ERA5-Land data (provided by the European Centre for Medium-

Range Weather Forecasts, ECMWF) are released with a five-day delay, their direct use in operational real-time LEWSs is constrained. Indeed, such a latency issues affects a broader range of Earth observation datasets. For instance, the NASA GPM (Global Precipitation Measurement) mission provides datasets with varying latencies, between the observation time and the publication one, varying from 4 hours to 3.5 months (Huffman et al., 2019). Similarly, NASA's SMAP (Soil Moisture Active Passive) soil moisture retrievals typically have a latency of 1–3 days, with latency increasing for higher-level or gap-filled products (Dashtian et al., 2024). Against this backdrop, the present study investigates the feasibility of using specifically delayed ERA5-Land soil moisture data for real-time landslide forecasting, recognizing that data latency is a common challenge shared by other sources potentially valuable for landslide prediction studies. Within this framework, we specifically assess the extent to which different publication lags (up to 15 days) affect the performance of ANN-based landslide prediction. Indeed, although the current latency of ERA5-Land soil moisture data is approximately 5 days, this study explores a broader latency range to evaluate the sensitivity of model performance to delayed information. This choice serves a dual purpose: first, to account for potential improvements in future reanalysis products, e.g., ERA6 (https://climate.copernicus.eu/sites/default/files/2022-09/S3_Hans_Hersbach_v1.pdf) as well as temporary disruptions in data availability due to maintenance or access issues; second, to investigate how the timeliness of soil moisture data affects landslide prediction capability. Understanding the extent to which performance decreases as latency increases allows us to better define the operational value of reanalysis data and identify time thresholds within which delayed soil moisture information remains effective for early warning applications. The models are evaluated using receiver operating characteristic (ROC) statistics, and the methodology is tested on the island of Sicily (Italy).

## 2 Material and methods

### 2.1 Overview and dataset creation

The proposed methodology is briefly schematized in in Fig. 1.

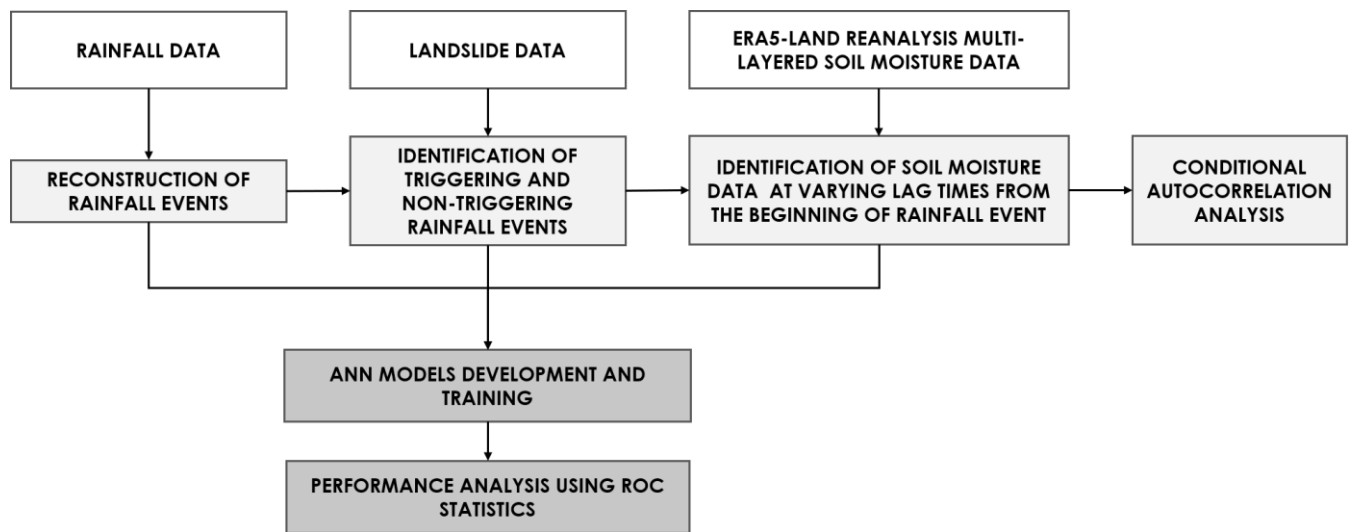

**Figure 1: Scheme of the methodology applied to explore the potential use of delayed ERA5-Land soil moisture products for the identification of landslide triggering conditions**

As illustrated, the first step of the methodology involves collecting rainfall and landslide data. Specifically, the FraneItalia inventory (Calvello and Pecoraro, 2018) is employed to gather information on observed landslides. This inventory records historical landslides throughout Italy since 2010, encompassing both fatal and non-fatal events. This detailed dataset enables the identification and selection of landslide events whose main cause is intense and/or prolonged rainfall. The CTRL-T (Calculation of Thresholds for Rainfall-Induced Landslides-Tool) code (Melillo et al., 2018) has then been applied to identify rainfall events most likely responsible for the observed slope failures. Briefly, the tool comprises different modules, each designed for a specific function. One of these modules reconstructs rainfall events in terms of duration $D$ (h) and cumulative event rainfall $E$ (mm) by analyzing a continuous rainfall time series, along with various climate and spatial parameters. Another module, on the other hand, selects the most representative rain gauge for each landslide event. This tool allows, thus, the reconstruction of both triggering and non-triggering rainfall events. On the other side, ERA5-Land reanalysis dataset provides soil moisture values ($\vartheta$ [$m^3\,m^{-3}$]) at four distinct depths (i.e., 0–7 cm, 7–28 cm, 28–100 cm, and 100–289 cm), at the hourly scale and at a high spatial resolution. In general terms, ERA5-Land is designed to improve the representation of land surface processes. Its land surface module is natively produced at ~9 km resolution, whereas the atmospheric forcing fields (e.g., precipitation, temperature) are statistically downscaled from the coarser ERA5 reanalysis (31 km) (Hersbach et al., 2020). This structure ensures consistency with ERA5 while enhancing the detail of land surface variables, making ERA5-Land particularly suitable for hydrological and landslide applications. Thus, multi-layered soil moisture data, from the ERA5-Land reanalysis project, at the onset of all rainfall events are then retrieved, together with the preceding values up to 15 days before. Regarding landslide information, in the present study, solely the FraneItalia inventory has been employed, in order to objectively compare the performance of the models using delayed soil moisture with our previous study utilizing non-delayed soil moisture (Distefano et al., 2023). Nonetheless, the proposed approach is fully

transferable to other landslide datasets, such as ITALICA (ITAlian rainfall-induced LandslIdes CAtalogue) (Peruccacci et
al., 2023), as far as the necessary information is available.

## 2.2 Preliminary correlation analysis

An exploratory analysis is firstly conducted, in order to understand from a purely statistical standpoint, how delay affects the
information content of soil moisture data from the ERA5-Land Reanalysis. To this aim, we computed a conditional
autocorrelation, i.e. the autocorrelation between the soil moisture at the beginning of triggering and non-triggering rainfall
events and its values in the preceding days, with time lags ranging from 0 to 15 days with respect to the rainfall event's
timing. Specifically, for each cell corresponding to a rainfall station soil moisture at each depth and at onset of rainfall events
is compared with values from preceding days. With reference to the generic cell corresponding to the rainfall station, let $S_{t(i)}$
with $i=1, 2,…,N$ represent the soil moisture at the onset time $t(i)$ of the generic rainfall event i and $k$ the lag time (in days).
We have then computed the linear correlation coefficient $\rho(k)$ between soil moisture values at lag $k$ and lag 0 as follows (Eq.
1):

$$\rho(k) = \frac{\sum_{i=1}^{N}(S_{t(i)}-\bar{S})(S_{t(i)-k}-\bar{S})}{\sqrt{\sum_{i=1}^{N}(S_{t(i)}-\bar{S})^2 \sum_{i=1}^{N}(S_{t(i)-k}-\bar{S})^2}} \tag{1}$$

where $S_{t(i)-k}$ denotes soil moisture lagged by $k$ days relative to the onset time $t(i)$ and $\bar{S}$ is the mean of $S_{t(i)-k}$. Eq. (1)
differs from the so-called autocorrelation for being conditioned to the beginning time of rainfall events. For this reason, we
refer to $\rho(k)$ as a conditional autocorrelation. Eq. 1 has been applied with reference to all rainfall events, as well as to the
triggering and non-triggering events.

## 2.3 Artificial Neural Network models (ANNs)

ANNs are a class of machine learning models designed to mimic the structure and functioning of neural systems, particularly
how the human brain processes information (Haykin, 1994). They consist of numerous interconnected processing elements
(i.e., neurons) that work together to solve complex tasks. Similar to human learning, ANNs improve their performance
through experience, adjusting the strength of connections between neurons (synapses) to improve their ability to recognize
patterns and make predictions (Grosan and Abraham, 2011; Haykin, 1994). In this study, the following input variables were
considered: (i) precipitation duration $D$, (ii) cumulative precipitation $E$, (iii) volumetric soil water content $S$ at different time-
lag relatively to the initial instant of rainfall events. Regarding the lagged soil water content, we have chosen to consider the
data relative only to a given lag. In other words, the ANNs for a given lag of $k$ days consider as input soil moisture data only
for that lag, and not for larger lags. All input variables considered have been transformed to their natural logarithms – this
has been empirically proven to improve the efficiency of ANN calibration (Distefano et al., 2022). Fig. 2 shows the
architecture of the implemented ANN model and the input variables considered in this study.

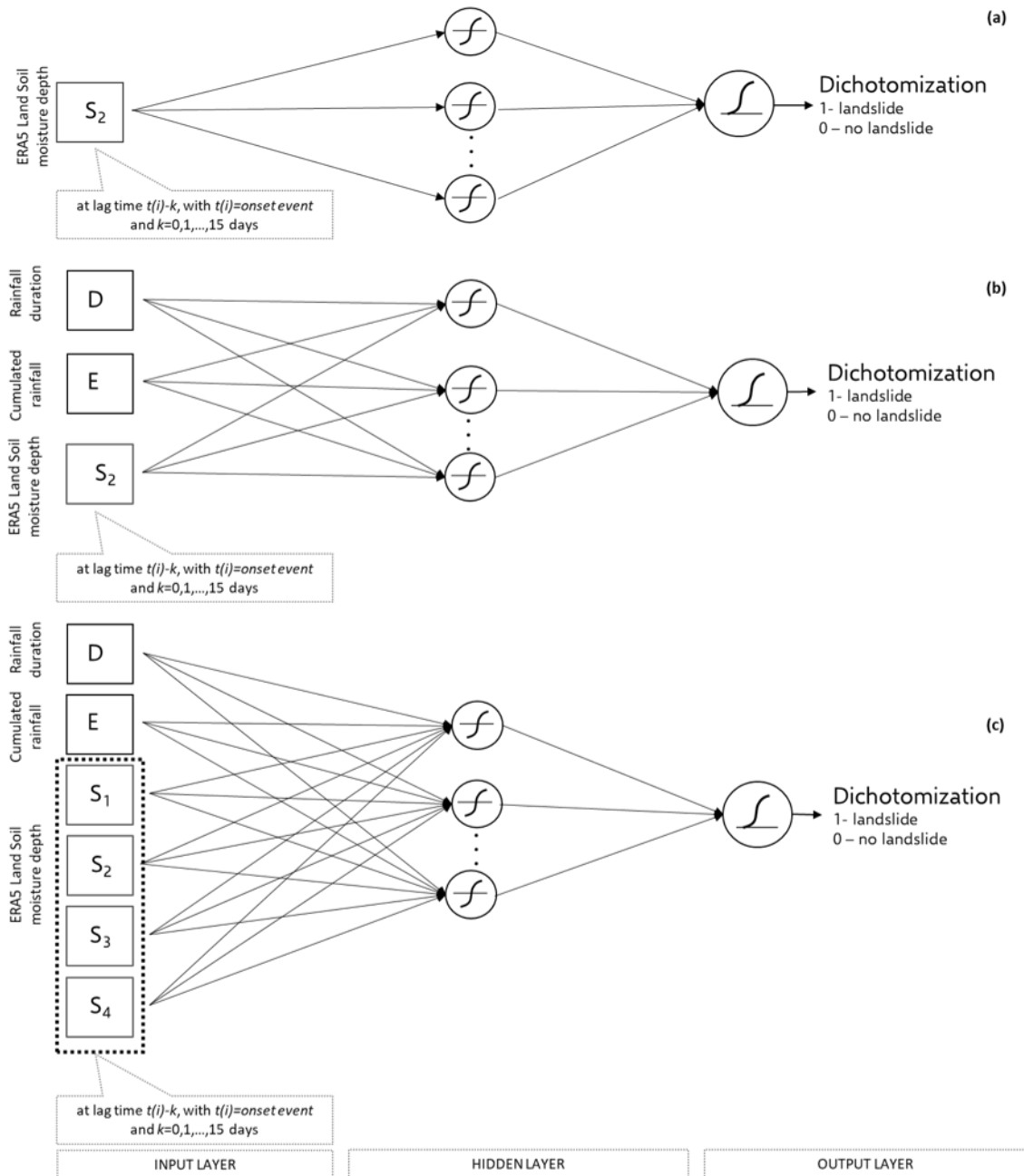

Figure 2: Sketch of the implemented ANN model. Soil moisture subscripts (*S*) indicate the soil layer (depth range) of the ERA5-Land reanalysis dataset. (a) Soil moisture in the 7–28 cm layer at increasing lags; (b) Soil moisture in the 7–28 cm layer at increasing lags together with cumulated rainfall and rainfall duration; (c) Multilayer soil moisture at increasing lags together with cumulated rainfall and rainfall duration. All variables are used as inputs to the ANN

The ANN models were developed using the MATLAB® Deep Learning Toolbox. Training was performed with the scaled conjugate gradient backpropagation algorithm (Møller, 1993), and model performance was evaluated using the cross-entropy loss function (Kline and Berardi, 2005), which is widely adopted in classification tasks and is known to be robust to class imbalance. This ensures a reliable training process even under unbalanced conditions, where the number of reconstructed non-triggering rainfall events could exceed that of triggering events by multiple orders of magnitude. Indeed, within a study currently under preparation, we have investigated the influence of different proportions of non-triggering and triggering events. The outcome of these tests suggests that it not necessary to balance the datasets with our model setting and that even unbalanced datasets lead to more robust models, as, while using the entire information available, they reflect the actual likelihood of triggering and non-triggering events. Hence, the entire dataset, comprising both triggering and non-triggering rainfall events, was randomly split into three subsets: 70% for training, 15% for validation, and 15% for testing. The allocation was performed randomly according to a uniform distribution over the dataset indices, meaning that each observation had the same probability of being assigned to any of the three subsets. To account for the variability introduced by this random sampling process, each neural network configuration was trained and evaluated 30 times, each with a different random split. This approach allows for a robust assessment of model performance and simulates realistic operational conditions, where landslide-triggering events are relatively rare. Furthermore, for each of these training sessions the number of neurons has been varied from 5 to 20, choosing at each session the number that corresponds to the highest performances.

The following simulations using different input datasets: (i) soil moisture at the second-layer depth only; (ii) rainfall duration ($D$, hours), rainfall depth ($E$, mm), and soil moisture at the second-layer depth; and (iii) rainfall duration ($D$), rainfall depth ($E$), and soil moisture at all depth layers. These input combinations were selected based on the findings of Distefano et al. (2023), who reported that the highest performance was achieved when considering soil moisture from the second layer, with only slightly higher performances with all four layers.

To assess the accuracy of the ANNs' output, hence the performance of the model's prediction, ROCs (Receiver Operating Characteristics) analysis is carried out. Specifically, the True Skill Statistic (TSS) index is computed for each simulation, corresponding to every investigated combination of input variables. TSS is a measure of how much better a model performs compared to random guessing, making it a reliable measure of the model's ability, i.e., ANNs, to correctly classify both positive and negative cases, namely triggering and non-triggering rainfall events. Its computation is based on the confusion matrix which provides a detailed breakdown of the model's predictions compared to actual outcomes (Fawcett, 2006). It is a square matrix where rows represent the actual classes and columns represent the predicted classes. Thus, for a binary classification problem, like the discrimination between triggering and non- triggering conditions, the matrix distinguishes: i) True Positives (TP) as triggering events correctly identified as triggering events; ii) True Negatives (TN) as non-triggering events correctly identified as non-triggering events; iii) False Positives (FP) as non-triggering events incorrectly classified as triggering events (false alarms); and iv) False Negatives (FN) as triggering events incorrectly classified as non-triggering events (missed alarms). Therefore, TSS (Eq. 4) index is defined as the difference between the True Positive Rate or

Sensitivity (TPR) (Eq. 2) and the False Positive Rate (FPR) (Eq. 3). The highest performances correspond to TSS=1 when the model correctly predicts for all the analysed data.

$$\text{TPR} = \frac{\text{TP}}{(\text{TP+FN})} \tag{2}$$

$$\text{FPR} = \frac{\text{FP}}{(\text{TN+FP})} \tag{3}$$

$$\text{TSS} = \text{TPR} - \text{FPR} \tag{4}$$

Thus, the network outputs, originally ranging from 0 to 1, were converted into binary predictions through a thresholding procedure, whereby an event is classified as a landslide (i.e., output = 1) if the model output exceeds a predefined threshold, and as a non-triggering event otherwise. This threshold was selected by maximizing the TSS, which ensures an optimal trade-off between TPR and FPR. While TSS is widely adopted in the literature for evaluating binary classifiers (Frattini et al., 2010), other metrics can also be employed to assess model performance and complexity. For example, information criteria such as the Akaike Information Criterion (AIC) and the Bayesian Information Criterion (BIC) have been effectively applied in studies comparing heterogeneous or hybrid models (Dutta et al., 2025; Patton et al., 2023; Quraishi and Choudhury, 2023) to balance model fit and parametrization. In the case of neural networks, however, the most common strategies rely on the use of independent training, validation, and test datasets, combined with early stopping, which—as in the present study—effectively control overfitting and ensure generalization.

## 3 Study area and data

Sicily (Southern Italy) is selected as study area to perform our analyses (Fig. 3). Annual rainfall in the region ranges on average from 700 mm to 800 mm, with the majority occurring during the autumn and winter months. Red triangles, within the study area map, represent the 207 landslides retrieved by FraneItalia database (Calvello and Pecoraro, 2018) from 2010 to 2018 along with their longitude–latitude coordinates (WGS84 datum) and the triggering time information, if the latter is available (otherwise, it is assumed at the end of the day). Furthermore, the mentioned landslide database distinguishes between two categories of events: single landslide events (SLEs) and areal landslide events (ALEs). SLEs generally provide more precise temporal and spatial information on the failure, whereas ALEs consist of multiple landslides within a defined area and are typically associated with lower spatial resolution—often limited to administrative units such as municipalities. Despite the potential uncertainties in the location and data for both SLEs and ALEs, we have chosen to retain all events in order to have statistically representative subsamples for training, validating and testing the ANNs. More specifically, only those landslides linked to rainfall as the main triggering factor were considered. This includes events where the mobilized material was consistent with rainfall-induced landslides, while other types (e.g., rockfalls) were excluded. The final dataset consisted of 207 rainfall-related landslides. The CTRL-T software enabled the reconstruction of triggering conditions for 144

of these events. Among them, movement type information is unavailable for 126 events (87.5%). Of the remaining cases, 10 were identified as areal rockfalls (6.9%), and both flows and slides accounted for 4 events each (5.6%). Regarding temporal resolution, for 103 landslides only the day of occurrence was known, while for the remaining cases more detailed information—such as the hour or time of the day (e.g., morning, afternoon, or evening)—was available. For events with daily resolution, the failure was assumed to occur at the end of the day. When more precise timing was reported, the failure was assumed to coincide with the peak rainfall within the reported time window. However, as discussed by Peres et al. (2018), even small shifts in the assumed timing of landslide occurrence—whether anticipating or delaying the event—can affect the estimation of rainfall thresholds. Despite this limitation, the lack of precise temporal information cannot currently be resolved. Nonetheless, the consistency of the dataset across all models ensures an objective comparison of their performance.

The map also shows the ERA5-Land grid at which extent soil moisture data are released. Specifically, ERA5-Land reanalysis data are released with a horizontal resolution of 0.1° x 0.1° corresponding to about 9 x 9 km grid at the latitudes of the case study area (https://cds.climate.copernicus.eu/datasets/reanalysis-era5-land?tab=download).

Green circles, instead, represent the locations of the 306 gauges installed in Sicily and managed by the three main Sicilian gauging network, namely the Regional Water Observatory (Osservatorio delle Acque, OdA), the Sicilian Agrometeorological Information Service (Servizio Informativo Agrometeorologico Siciliano, SIAS), and the Regional Department of Civil Protection (Dipartimento Regionale di Protezione Civile, DRPC). Rainfall data are, thus, provided at the hourly scale from 2009 up to 2018 (Distefano et al., 2022).

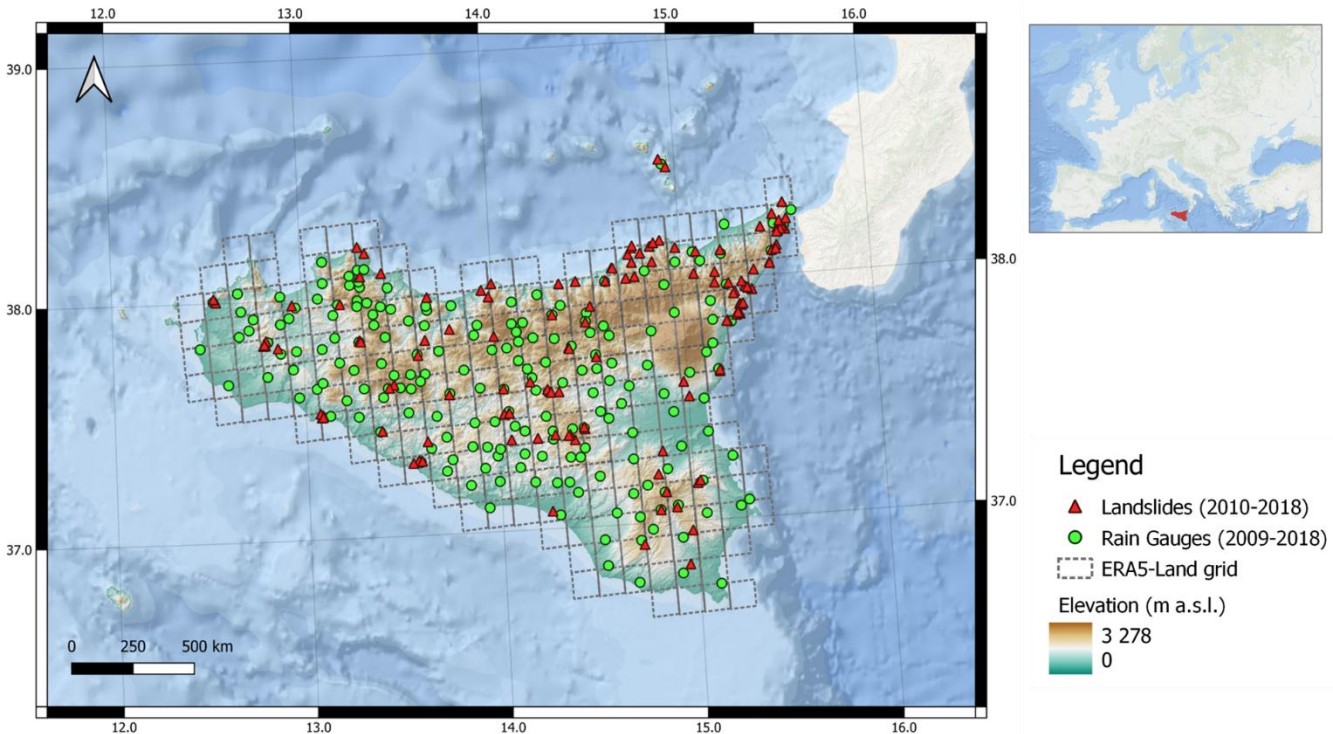

225

**Figure 3: Study area map ((credit to http://www.sinanet.isprambiente.it/it/sia-ispra/download, last access: 01 April 2025, and ESRI, 2020). Red triangles represent the locations of the occurred landslides in the study area from 2010 to 2018 and retrieved by the FraneItalia catalogue; green dots represent the locations of the 306 gauges installed in Sicily.**

As mentioned in the previous section, CTRL-T tool enables the identification of both triggering and non-triggering rainfall events for each rainfall station based the hourly rainfall, and by setting a combination of regional parameters. These parameters are designed to account for climatic seasonality, namely at defining the warm period $C_W$ and the cold one $C_C$; the length of dry period exceeded which an hourly measurement is considered isolated and thus removed ($P_1$); the time periods used to remove irrelevant amounts of rainfall between rainfall sub-events ($P_2$), and ($P_3$); and the minimum dry period separating two rainfall events assumed to be single or multiple sub-events ($P_4$). Moreover, even the instrumental sensitivity of a rain gauge ($G_S$), as well the radius of the buffer to assign each landslide to the closest rain gauge ($R_B$), and the minimum value $E_R$ above which a rainfall measurement is classified as isolated can be set. For further details on these parameters, readers are referred to Melillo et al. (2018). Table 1 summarize the parameters' values adopted for our analysis. Specifically, the $R_B$ value of 16 km was used; however, since the mean distance between rain gauges and landslides was approximately 5 km, this maximum value was seldom reached. The warm season CW was defined as the period from April to October, and the cold season CC as the period from November to March. Accordingly, parameters *sws* and *ews*, indicating the beginning of the warm season the end of the warm season, are set equal to 4 (i.e., April) and 10 (i.e., October), respectively. These values are consistent with those proposed by Melillo et al., (2015).

Table 1: CTRL-T parameters for the reconstruction of the rainfall events used in the present study (after Distefano et al., 2022).

| $G_S$ (mm) | $E_R$ (mm) | $R_B$ (km) | sws (-) | ews (-) | $P_1$ (h) | | $P_2$ (h) | | $P_3$ (h) | | $P_4$ (h) | |
|---|---|---|---|---|---|---|---|---|---|---|---|---|
| | | | | | $C_w$ | $C_c$ | $C_w$ | $C_c$ | $C_w$ | $C_c$ | $C_w$ | $C_c$ |
| 0.2 | 0.2 | 16 | 4 | 10 | 3 | 6 | 6 | 12 | 1 | 1 | 48 | 96 |

## 4 Results and discussion

First, the explanatory analysis regarding conditional autocorrelation analysis has been carried out, in order to understand how information content may decrease with lag from a statistical standpoint. For all four soil layers, the conditional autocorrelation function has a decreasing trend, even though for the deepest layer the decay is minimal (Fig. 4).

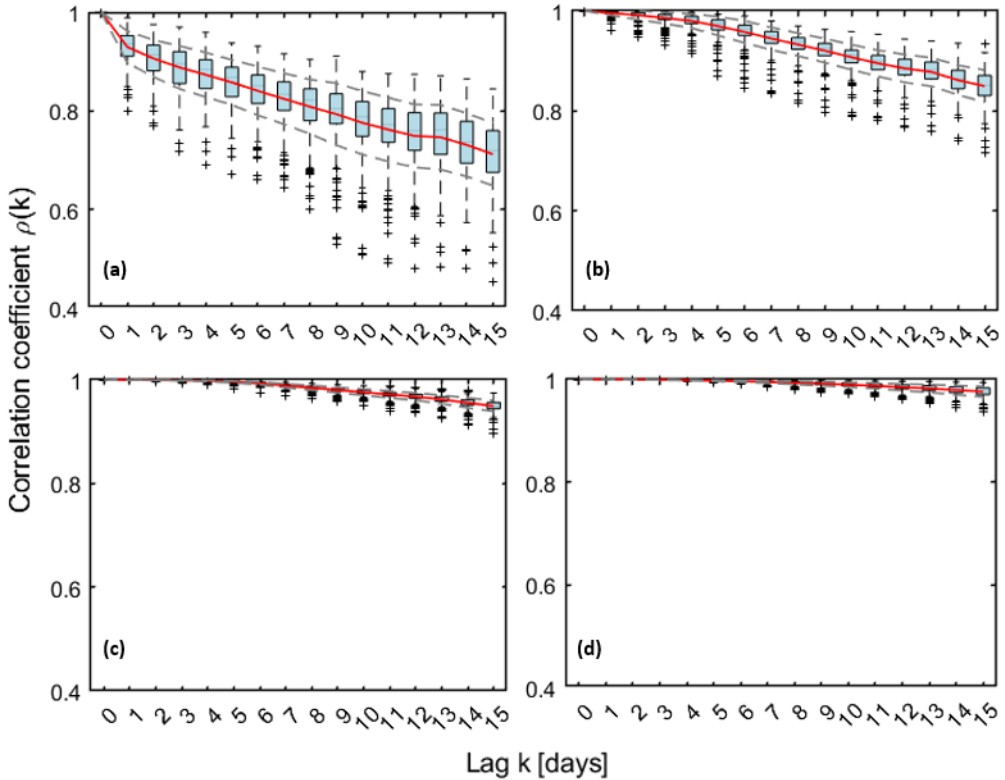

**Figure 4: Conditional autocorrelation function for soil moisture at the four ERA5-Land depths (time lags from 1 to 15 days) (a) 0-7 cm; (b) 7-28 cm; (c) 28-100 cm; (d) 100-239 cm.**

The shallowest layers, particularly the first one, appear to be the most responsive to the effect of lag, showing a notable drop in conditional autocorrelation as early as lag = 1. This is consistent with the fact that shallow layers are mostly sensitive to the forcing of even small rainfall events that induce high frequency fluctuations in soil moisture, with a consequent reduction of short short-term memory in the process. As deeper layers are considered, high frequency forcings tend to be dampened by

the presence of the above layers resulting in a more stable pattern in time and stronger autocorrelation, at least for the investigated lags. For subsequent analyses, as outlined in the previous sections, we focused on layer 2 (7–28 cm), which had already been identified in Distefano et al. (2023). This because, in the same study, a systematic evaluation of alternative input combinations (D, E, $S_1$, $S_2$, $S_3$, $S_4$, and their subsets) indicates that soil moisture at the second layer depth consistently provided the best model performance.

Moving to the results of ANNs for the prediction of landslide triggering conditions, starting with those using only soil moisture data without precipitation (Table 2), a TSS of 0.53 is achieved when no lag is considered. This performance is slightly higher than that obtained with a traditional approach (i.e., the power-law threshold based on rainfall intensity and duration, which has a TSS of 0.50; (Palazzolo et al., 2023)) but slightly lower than that achieved by Distefano et al. (2023), where an ANN model using rainfall intensity and duration lead to significantly higher performances (TSS = 0.59). When a 5-day lag is considered, TSS drops further to 0.43.

**Table 2: Performances of ANNs using exclusively soil moisture at the second layer (7-28 cm).**

| Lag [days] | Hidden neurons [mean] | TSS$_{all}$ (mean) | TSS$_{train}$ (mean) | TSS$_{val}$ (mean) | TSS$_{test}$ (mean) | TPR$_{all}$ (mean) | FPR$_{all}$ (mean) |
|---|---|---|---|---|---|---|---|
| 0 | 9.17 | **0.53** | 0.57 | 0.50 | 0.48 | **0.74** | **0.21** |
| 1 | 5.50 | 0.48 | 0.52 | 0.45 | 0.45 | 0.72 | 0.24 |
| 2 | 12.40 | 0.44 | 0.48 | 0.42 | 0.42 | 0.73 | 0.29 |
| 3 | 15.40 | 0.44 | 0.47 | 0.42 | 0.42 | 0.78 | 0.33 |
| 4 | 15.60 | 0.44 | 0.47 | 0.41 | 0.41 | 0.78 | 0.35 |
| 5 | 6.20 | **0.43** | 0.46 | 0.39 | 0.39 | **0.71** | **0.28** |
| 6 | 8.93 | 0.42 | 0.46 | 0.38 | 0.38 | 0.71 | 0.29 |
| 7 | 11.30 | 0.40 | 0.44 | 0.36 | 0.38 | 0.68 | 0.28 |
| 8 | 13.53 | 0.39 | 0.44 | 0.35 | 0.34 | 0.64 | 0.25 |
| 9 | 17.70 | 0.41 | 0.44 | 0.36 | 0.38 | 0.66 | 0.26 |
| 10 | 13.87 | 0.38 | 0.42 | 0.34 | 0.35 | 0.64 | 0.27 |
| 11 | 9.27 | 0.36 | 0.40 | 0.33 | 0.32 | 0.61 | 0.25 |
| 12 | 10.13 | 0.35 | 0.38 | 0.33 | 0.32 | 0.73 | 0.38 |
| 13 | 12.00 | 0.34 | 0.39 | 0.34 | 0.31 | 0.70 | 0.36 |
| 14 | 11.57 | 0.35 | 0.39 | 0.32 | 0.33 | 0.73 | 0.38 |
| 15 | 11.73 | 0.36 | 0.40 | 0.32 | 0.34 | 0.72 | 0.36 |

As expected, including rainfall duration (D (h)) and rainfall depth (E (mm)) besides soil moisture leads to a significant improvement in performance (Fig. 5). In more detail, when D-E-$S_2$ dataset is considered as input to the model, a TSS of 0.76 is achieved, while incorporating the entire D-E-$S_{all}$ dataset increases the TSS to 0.78. Nevertheless, in both cases,

performance reduces as the time-lag increases. Focusing on a 5-day time delay a slight drop in performance is observed, with TSS values of 0.70 and 0.72, respectively. Although these values are lower, they are still above the ANN model based solely on rainfall data. As the time delay increases, performance continues to marginally decrease, reaching a TSS of around 0.68

at the 15-day time lag in both analyzed scenarios, namely a mean decrease of just 0.09 point.

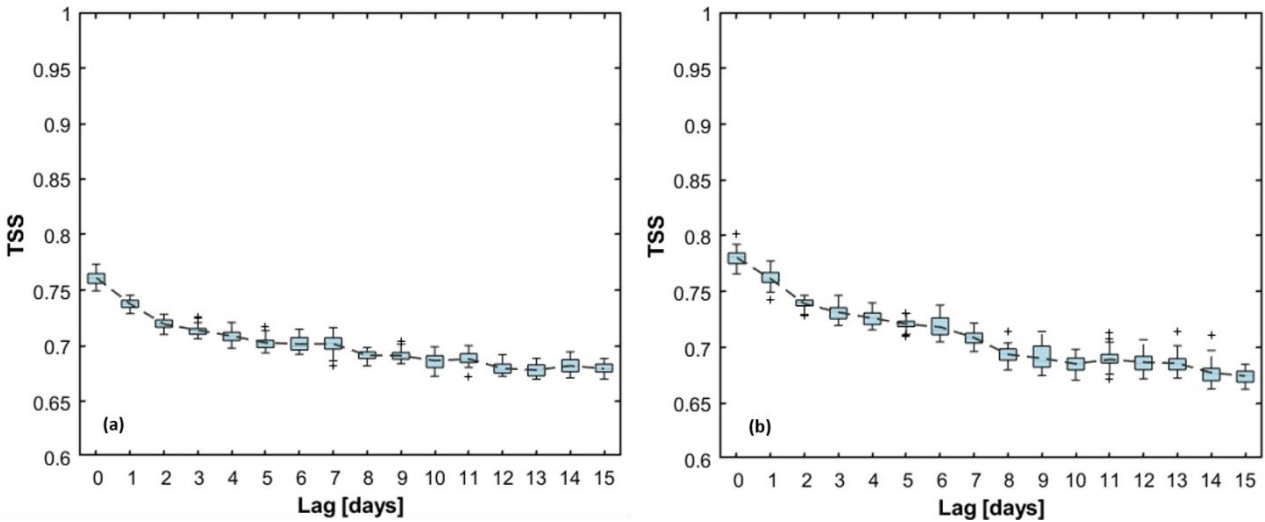

**Figure 5***: **TSS values from 30 ANN training runs, with the dashed grey line representing the mean TSS across lag time. (a) simulation set with rainfall duration (D, [hours]), rainfall depth (E, [mm]) and soil moisture at the second-layer depth as input dataset; (b) simulation set with rainfall duration (D, [hours]), rainfall depth (E, [mm]) and soil moisture at all -layer depth as input**
**dataset.**

Overall, box-plots present a narrow interquartile range, supporting the adequacy of the input dataset and confirming robustness of the results. Their width further depicts how sensitive the model performance is to the random selection of subsets, with narrow boxes indicating stable outcomes across different splits. More details about simulations carried out are summarized in Tables 3 and 4. For the E-D-S2 configuration (Table 3), short delays up to 5 days determine a modest

reduction in TSS, with TPR almost unchanged (TPR from 0.89 to 0.87) and a moderate increase in the false alarms rate (FPR from 0.12 to 0.16). At longer delays, performance stabilized around TSS = 0.68–0.70, still with relatively high TPR but with higher false alarm rates (up to 0.20). When soil moisture from all four layers was included (Table 4), performances improve slightly systematically; but for instance, with lags up to 5 days, TPR is maintained (TPR = 0.89–0.90), and FPR increases only slightly (up to 0.18). Beyond 5 days, performance gradually declines (TSS = 0.67–0.69, FPR ≈ 0.20–0.21). Compared

to the single-layer setup, the multilayer configuration consistently yielded higher TSS values and lower FPRs, confirming that the vertical soil moisture profile provides a more realistic representation of antecedent hydrological conditions. Nevertheless, the increase in performance respect to the most representative layer may not be worth the increase of input complexity.

**Table 3: Performances of ANNs using rainfall duration, depth and soil moisture at the second layer (7-28 cm).**

| Lag (days) | Hidden neurons (mean) | TSS$_{all}$ (mean) | TSS$_{train}$ (mean) | TSS$_{val}$ (mean) | TSS$_{test}$ (mean) | TPR$_{all}$ (mean) | FPR$_{all}$ (mean) |
|---|---|---|---|---|---|---|---|
| 0 | 13.23 | **0.76** | 0.78 | 0.73 | 0.73 | **0.89** | **0.12** |
| 1 | 13.33 | 0.74 | 0.76 | 0.71 | 0.71 | 0.88 | 0.15 |
| 2 | 14.83 | 0.72 | 0.74 | 0.69 | 0.70 | 0.88 | 0.16 |
| 3 | 13.33 | 0.71 | 0.74 | 0.69 | 0.68 | 0.88 | 0.17 |
| 4 | 14.37 | 0.71 | 0.74 | 0.69 | 0.68 | 0.88 | 0.17 |
| 5 | 14.30 | **0.70** | 0.73 | 0.68 | 0.67 | **0.87** | **0.16** |
| 6 | 14.60 | 0.70 | 0.73 | 0.69 | 0.67 | 0.88 | 0.18 |
| 7 | 15.03 | 0.70 | 0.73 | 0.67 | 0.68 | 0.88 | 0.18 |
| 8 | 15.20 | 0.69 | 0.72 | 0.67 | 0.66 | 0.87 | 0.18 |
| 9 | 13.63 | 0.69 | 0.72 | 0.67 | 0.65 | 0.85 | 0.16 |
| 10 | 15.10 | 0.69 | 0.71 | 0.67 | 0.66 | 0.85 | 0.16 |
| 11 | 15.63 | 0.69 | 0.72 | 0.67 | 0.65 | 0.86 | 0.17 |
| 12 | 14.53 | 0.68 | 0.71 | 0.65 | 0.64 | 0.85 | 0.17 |
| 13 | 15.07 | 0.68 | 0.71 | 0.64 | 0.67 | 0.87 | 0.19 |
| 14 | 16.60 | 0.68 | 0.71 | 0.66 | 0.66 | 0.88 | 0.20 |
| 15 | 15.10 | 0.68 | 0.71 | 0.66 | 0.65 | 0.86 | 0.18 |

**Table 4*:* Performances of ANNs using rainfall duration, depth and soil moisture at all four layers.**

| Lag (days) | Hidden neurons (mean) | TSS$_{all}$ (mean) | TSS$_{train}$ (mean) | TSS$_{val}$ (mean) | TSS$_{test}$ (mean) | TPR$_{all}$ (mean) | FPR$_{all}$ (mean) |
|---|---|---|---|---|---|---|---|
| 0 | 15.20 | **0.78** | 0.81 | 0.76 | 0.75 | **0.89** | **0.11** |
| 1 | 14.07 | 0.76 | 0.79 | 0.74 | 0.74 | 0.89 | 0.13 |
| 2 | 15.53 | 0.74 | 0.77 | 0.72 | 0.72 | 0.90 | 0.16 |
| 3 | 16.80 | 0.73 | 0.76 | 0.71 | 0.70 | 0.89 | 0.15 |
| 4 | 15.47 | 0.73 | 0.75 | 0.71 | 0.70 | 0.89 | 0.17 |
| 5 | 13.93 | **0.72** | 0.75 | 0.71 | 0.70 | **0.90** | **0.18** |
| 6 | 14.60 | 0.72 | 0.74 | 0.70 | 0.69 | 0.89 | 0.17 |
| 7 | 16.30 | 0.71 | 0.74 | 0.69 | 0.67 | 0.88 | 0.17 |
| 8 | 15.10 | 0.69 | 0.72 | 0.67 | 0.67 | 0.86 | 0.16 |
| 9 | 15.60 | 0.69 | 0.72 | 0.67 | 0.67 | 0.85 | 0.16 |
| 10 | 14.37 | 0.69 | 0.71 | 0.68 | 0.67 | 0.85 | 0.16 |

| 11 | 15.33 | 0.69 | 0.72 | 0.68 | 0.66 | 0.85 | 0.16 |
| 12 | 14.30 | 0.69 | 0.72 | 0.66 | 0.66 | 0.85 | 0.17 |
| 13 | 14.60 | 0.69 | 0.71 | 0.67 | 0.66 | 0.88 | 0.20 |
| 14 | 16.30 | 0.68 | 0.71 | 0.66 | 0.65 | 0.89 | 0.21 |
| 15 | 15.23 | 0.67 | 0.70 | 0.65 | 0.65 | 0.88 | 0.20 |

In these latter two simulation sets, the results indicate that, on average, using 15 neurons provides the best performance, providing a suitable balance between model complexity and generalization capabilities.

Notably, a lower number of neurons may lead to underfitting, limiting the network's capacity to capture the patterns in the data. Conversely, increasing the neuron beyond 15 does not yield significant improvements and may introduce unnecessary complexity. In this regard, looking at the simulation set exclusively using $S_2$ as input, the number of hidden neurons decreases at 9, on average, reasonably due to the lower complexity of the input dataset.

As a whole, results show that delayed ERA5-Land soil moisture information can be considered a useful predictor for landslide prediction, although the performance of the prediction model slightly decreases as the time-lag increases. Indeed, the improved performance, even with lagged data, can be explained by the slow variation of soil moisture during dry periods preceding rainfall events. Consequently, soil moisture at the onset of a rainfall event does not differ significantly from that recorded five or more days earlier. Our results corroborate the findings of Marino et al. (2020), where, based on Monte Carlo simulations, they have shown that soil moisture estimates, even when derived from a simplified hydrological model—rather than satellite or in-situ data—can significantly enhance the robustness of forecasting thresholds under different uncertainty scenarios by reducing false alarms. These findings emphasize that high-accuracy, point-scale observations, which have proven to have more information content than satellite data (Mirus et al., 2025), are not strictly necessary to extract meaningful predictive signals. On the contrary, large-scale products, when properly interpreted, can effectively reflect the temporal dynamics of soil moisture that control slope stability. Therefore, satellite or reanalysis-based soil moisture datasets represent a valuable resource to support landslide early warning systems, especially in areas lacking soil moisture monitoring networks, as in Sicily, where there are no publicly available datasets – for instance, the closest sensors from the International Soil Moisture Network (https://ismn.earth/en/) are in Calabria. This is particularly relevant in contexts where current warning procedures rely solely on rainfall thresholds, as the inclusion of soil moisture, even coarse or delayed, provides useful information on catchment predisposition and can improve the reliability of alerts.

Concerning, instead, the generalizability and broader applicability of the proposed approach, it is worth noting that the rainfall events considered for model development covered a wide range of characteristics, with durations between 1 hour and 10 days and cumulative totals up to 300 mm (Distefano, 2023). In addition, soil moisture values ranged between 0.15 and 0.45, effectively spanning the porosity domain of most common soil types. These conditions delineate the applicability domain of the trained models and support their potential transferability to different climate and geological conditions.

**5 Conclusions**

This study proposed an ANN-based framework to evaluate whether ERA5-Land reanalysis soil moisture data can effectively serve as real-time data for landslide prediction. Building on the work of Distefano et al. (2023), our investigation focused on the classification capabilities of ANNs to identify triggering conditions using rainfall duration, rainfall depth, and delayed multi-layered soil moisture information. Specifically, ERA5-Land data are released with an approximately 5-day delay, limiting their immediate availability for real-time landslide prediction. The analysis was conducted with a focus on Sicily,

Italy. Three simulation sets were examined, progressively increasing in complexity based on the number of variables considered, with performance evaluated in terms of TSS. Significant performance was achieved when rainfall duration, rainfall depth, and soil moisture information at the second layer depth were used as input into ANNs. In this case, results demonstrated that soil moisture lagged data did not significantly undermine landslide prediction performance, as the TSS decreased from 0.76 with lag=0 to 0.70 with a 5-day lag. Even with a lag of 15 days, the prediction performance of ANNs is

still significantly higher than those models using only rainfall event characteristics, corresponding to a decrease of just 0.08 points. When all available soil moisture information is used as input, together with rainfall depth and duration, the model reaches even slightly higher predictive performance, comparable with those obtained using the most representative soil moisture layer. However, it may not be worth the increase of input complexity. These findings underscore the potential of integrating ERA5-Land multi-layered soil moisture data into LEWSs, despite their delayed release. Indeed, their regular

availability in both time and space enhances their suitability as a reliable tool for these systems. Further developments of this study will consider the use of rainfall forecasts at various temporal horizons in order to assess effective the operational predictive capability of the investigated tools.

**Author contributions**

Conceptualization was done by D.J.P., A.C. and N.P.; formal analysis by D.J.P., A.C. and N.P.; investigation by D.J.P. and

345 N.P.; methodology by D.J.P. and N.P.; coding by D.J.P., N.P. and R.D.Z.; writing the original draft by D.J.P., A.C., N.P. and R.D.Z. All authors have read and agreed to the published version of the paper.

**Data availability**

The FraneItalia landslides catalog is available at https://doi.org/10.17632/zygb8jygrw.2 (Calvello and Pecoraro, 2018). Rainfall measurements are available at the website of the Servizio Informativo Agrometeorologico Siciliano (SIAS)

(http://www.sias.regione.sicilia.it/, SIAS, 2025) and at the Osservatorio delle Acque (http://www.bio.isprambiente.it/annalipdf/, ISPRA, 2025). Reanalysis soil moisture data are available from https://doi.org/10.24381/cds.e2161bac (Muñoz-Sabater et al., 2021)

## Acknowledgements

This work was supported by the PRIN 2022 project "ITALERT – Prediction of Rainfall-INduced landslides – Improving multi-scale TerritoriAL Early warning through aRTificial intelligence" – funded by the Italian Ministry of University and Research (CUP E53D23004150006). The authors would like to thank Valerio Santoro for his valuable input in the early stages of this work.

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
