# Peer review of "Use of delayed ERA5-Land soil moisture products for improving landslide early warning"

_EGUsphere, 2025_

## Referee Comment (RC2)

**GENERAL COMMENT**

The contribution "Use of delayed ERA5-Land soil moisture products for improving landslide early warning" by N. Palazzolo and co-Authors is interesting and addresses questions relevant with the scope of NHESS. In this paper the researchers conducted a series of direct experiments. They aimed to determine how effective it is to use antecedent soil moisture data (ranging from 0 to 15 days old) from a global reanalysis model, combined with rainfall data, to predict landslide triggers using machine learning. As one might expect, the findings indicate that this historical soil moisture data enhances model performance, though its benefit diminishes as the time lag increases. Results, and discussion sections are short compared to amount of work done. They should be increased. The theoretical background is well-argued but not complete. Review of literature seems completed. The description of study area is sufficiently complete. The description of methodology and successive parts of paper are well organized but not complete. The readability of the whole paper is good with a good English. In general, the synthetic approach of the study is very clear, but I think that going into more detail on several aspects (later reported) could be useful for the readers. It can be published on NHESS journal only after minor revision.

**SPECIFIC COMMENTS**

I have some specific comments that should be addressed before the manuscript can be accepted for publication in NHESS journal.

**1 Introduction (Line 30-40)**

My main comment regards the conceptual background of the comparing of the performance of different models. It would be necessary to better understand from the authors whether it is possible to directly compare the performance of the models with and without soil moisture (on the empirical rainfall thresholds) information based exclusively on the comparison with the TSS index or whether other strategies exist in this direction.

**2 Material and methods**

(Line 48-49) Have you tried using landslide data from other catalogues? (i.e. Peruccacci et al., 2023)

(Line 55-60) The Authors reconstruct triggering and non-triggering rainfall events.

I didn't understand the following list of things:

The rainfall event or landslide triggering condition (MPRC automatically reconstructed by the tool) normally represents a subset of the entire rainfall event while non-triggering rainfall events are always considered as a whole. Did the authors take this difference into account when creating the input data for the neural network? How many triggering conditions coincide with the entire rainfall event? If not, how many of them differ significantly in terms of duration and cumulative rainfall, and what justifies the choice of these two sets of triggered and non-triggered rainfall events? The authors should provide a better argument and commentary on the method used.

The number of reconstructed non-triggering rainfall events is at least two orders of magnitude greater than those that trigger landslides. What procedure was used to account for this substantial difference? Are the samples used for the training, validation and testing phases balanced or unbalanced? What observations are used in this last case, and how is sampling performed? Have statistical tests been conducted to confirm whether these samples are representative of the population? Please specify the setting more precisely.

**2.3 Artificial Neural Network models (ANNs)**

(Line 80-90) ANNs Input variables/data

The reconstruction of the rainfall conditions that triggered landslides is performed starting from the assumption that in the place where the landslide occurs the rainfall is the same as that measured on a representative rain gauge (specific criteria dependent on a parameterized variable that considers the distance was used by CTRL-T tool). As regards the choice of cells that contains information associated with the soil moisture status, it seems that the cell that includes the representative rain gauge chosen automatically by the tool has been considered.

Under this hypothesis (soil moisture associated with the cell that includes rain gauge) what considerations can be made from a physical point of view? Is it conceptually correct to look for a relationship between the saturation state of a soil different from that in which the triggering of a landslide occurs? If the cell that contains the landslide is considered, do the results change? Considering the spatial resolution of the ERA5-Land product (about 9km x 9km) probably in most cases the cell containing the landslide or the rain gauge is the same, but I believe that deepening this aspect by making descriptive statistics could be useful. Moreover, for an operational use, the forecast value returned as output by the model would describe the possibility that a landslide could happen in that specific cell for which the soil moisture value was provided as input. Therefore, in my opinion, it should perhaps also be used in the model's learning phase.

**3 Study area and data**

(Line 140-150) CTRL-T parameters setting

What is the length of the "warm" spring–summer period ($C_W$) and "cold" autumn–winter period ($C_C$). More in detail, what values were assigned to the variables sws (beginning of warm season) and ews (end of warm season), i.e. $C_W$ from May to October and $C_C$ from November to April correspond sws=5 (May), ews=10 (October). Please specify the setting more precisely.

Add the *sws* and *ews* parameters in the Table 1 (cfr. comment refered to Line 140-150)

**4 Results and discussion**

(Line 160-165)

Figure 4 shows box plots of the results for the autocorrelation conditions. The text provides a good description of the behaviour of the variations in terms of lag k. However, a visual analysis of the $\rho(k)$ values (k = 1, 2, ..., 15 days) in the different layer depth cases (b) 7–28 cm, (c) 28–100 cm, and (d) 100–239 cm reveals that the values are indistinguishable (perhaps due to the scale). Consequently, it is unclear why layer 2 was selected for subsequent analysis (b). Please explain better.

Line 175-190

How does performance change for all combinations of layer use? For example, what is the maximum TSS (mean) value for D-E-S1, D-E-S3, D-E-S4, D-E-S all-1, and so on? Has a systematic analysis been conducted to better understand the effect of each input data component introduced in network training? If it is not very time-consuming, I would suggest trying this approach.

**Figures description**

Description of Figure 2

In the structural diagram of the ANN, the symbol for dichotomy on the third layer is different from those for the three preceding nodes on layer 2.

I suggest to:

remove "---and---" because it's implicit in the figure

replace "depth" with "cumulated rainfall" (also extended to the text)

Specify the terms S (ERA5-Land depht)

Specify which is the input, hidden and output level.

Description of Figure 3

Add geographical grid and coordinate labels.

I suggest to change the colour filling of the peninsular part of Italy (i.e. the monochrome palette) because the scale is different respect to the study area.

I suggest increasing the size of the dots and changing the starting colour of palette from "blue" to "green" for greater contrast.

Add the description of the symbols in the caption.

Description of Figure 4

Formatting borders, label text, and ticks in black instead of grey

Add the description of the continuous red and the dashed grey curves in the caption (also extended to the text).

Remove the labels and headings from the x-axis for (a) and (b) and from the y-axis for (b) and (d). This should make the figure clearer by increasing its size.

Description of Figure 5

Formatting borders, label text, and ticks in black instead of grey

Change dimensions of '(a)' and '(b)' as in figure 4.

**TECHNICAL CORRECTIONS**

Below a list of some more detailed comments and suggestions referred to specific parts of the text.

(Line 54): replace "($D$, in hours)" with "$D$ (h)"

(Line 55): replace "($E$, in millimeters)" with "$E$ (mm)"

(Line 116): remove the space in "non-triggering"

(Line 120): remove the space in "(missed alarms)"

(Line 123): Change the 'T' term format

(Line 126): enter the comma in the range [0,1]

(Line 129): replace "southern Italy" with "Southern Italy"

(Line 31): replace "700" with "700 mm"

REFERENCES:

Peruccacci S., Gariano S.L., Melillo M., Solimano M., Guzzetti F., Brunetti M.T. (2023) The ITAlian rainfall-induced LandslIdes CAtalogue, an extensive and accurate spatio-temporal catalogue of rainfall-induced landslides in Italy. Earth Syst Sci Data, 15, 2863–2877, doi: 10.5194/essd-15-2863-2023

---

## Author Comment (AC1)

REFEREE'S COMMENTS ON egusphere-2025-1590

RC#1
**R: Thank you for the opportunity to review this short-form manuscript. The authors design a set of straightforward experiments that include testing the efficacy of using 0- to 15-day antecedent soil moisture information from a modeled global reanalysis data product, in conjunction with rainfall data, to identify the triggering conditions for landslides using machine learning. The Results and Conclusion are intuitive in that antecedent soil moisture improves model performance, with the benefit decreasing somewhat with increased lag. This deprecation in model performance seems minor for a lag that is equivalent to the latency of the modeled soil moisture data product (~5 days). Although I appreciate the streamlined presentation of this study, I think it would be helpful for readers to see more text related to (1) the kind of landslides this study is relevant to, (2) why the spatiotemporal resolution of the modeled soil moisture data product is appropriate for the landslide type(s) considered here, and (3) a deeper interpretation of the Results. Regarding #3 - What are the rainfall depth/duration characteristics and the antecedent soil moisture levels that the best-performing model settles on? And do these characteristics make sense relative to the landslide type(s) and/or any previously published regional thresholds? The objective of this study is crystal clear, but the authors may consider questions like these to expand the relevance of their work for the broader scientific community.**
**Sincerely,**
**Matthew A. Thomas**

A: Thank you for your general appreciation of our manuscript and your valuable comments. Here our replies to the issues raised:

1) Regarding the type of landslides our study is relevant to, we have limited the information in this manuscript as they are basically the same of our previous cited study (Distefano et al., 2023). Nevertheless, given also similar requests by other referees, more details may be added in the "Study area and data section". Text to add may be as follows:
   "*The landslide database distinguishes between two categories of events: single landslide events (SLEs) and areal landslide events (ALEs). Both types were incorporated in the reconstruction of the most comprehensive set of landslide occurrences. SLEs generally provide more precise temporal and spatial information on the failure, whereas ALEs consist of multiple landslides within a defined area and are typically associated with lower spatial resolution—often limited to administrative units such as municipalities. Despite the potential uncertainties in the location and data for both SLEs and ALEs, all events were retained for the threshold analysis, as excluding them would have led to an insufficient sample size for training, validating, and testing the neural network models. More specifically, only those landslides linked to rainfall as the triggering factor were considered. This includes events where the mobilized material was consistent with rainfall-induced landslides, while other types (e.g., rockfalls) were excluded. The final dataset consisted of 207 rainfall-related landslides. The CTRL-T software facilitated the reconstruction of triggering conditions for 144 of these events. Among them, movement type information is unavailable for 126 events (87.5%). Of the remaining cases, 10 were identified as areal rockfalls (6.9%), and both flows and slides accounted for 4 events each (5.6%). Regarding temporal resolution, for 103 landslides only the date of occurrence was known. For the rest, more specific information—such as the hour or time of day (e.g., morning, afternoon, or evening)—was available. In cases with daily resolution, the failure was assumed to have occurred at the end of the day, while in events with more detailed timing, the landslide was assigned to the time of peak rainfall.*"

2) Regarding the spatiotemporal resolution of the modeled soil moisture, the following details will be added to the revised manuscript in the section "Case Study and data":
   "*Regarding temporal resolution, we recognize the importance of temporal resolution in the context of landslide modeling. However, considering the typical precision and uncertainty associated with landslide timing information in most available datasets, we believe that an hourly temporal resolution is already sufficiently detailed for capturing the relevant triggering conditions in most cases. In fact, finer resolutions may not necessarily lead to a meaningful improvement in model performance, given the inherent limitations in the temporal accuracy of landslide occurrence records. With reference to spatial resolution, each cell of the ERA5 Land grid covers an area of about 9x9 = 81 km², which is consistent with the average density of available rain-*

*gauges, that corresponds to 1 rain-gauge per 84 km². This can provide a measure of a sufficient resolution for applying our proposed approach."*

3) Regarding a deeper interpretation of the results, i.e. more details on the rainfall depth/duration characteristics and the antecedent soil moisture levels that the best-performing model settles on, and whether these characteristics make sense relative to the landslide type(s) and/or any previously published regional thresholds, please consider the following. As highlighted by Distefano (2023) (see Fig. 7), the rainfall events used for the development of the neural networks exhibit durations ranging from 1 hour to 10 days and event rainfall totals up to 300 mm. These values define the applicability domain of the trained models. Nonetheless, it is worth noting that the same study reported even better predictive performance for a climatically distinct region in Norway, where event durations spanned from 24 hours to 40 days and rainfall accumulations reached approximately 400 mm. This suggests that the methodology may maintain robust performance even under differing climatic conditions, particularly when using lagged hydrometeorological inputs. Additionally, the soil moisture values included in the analysis range between 0.15 and 0.45, effectively covering the porosity spectrum of most soil types. This supports the generalizability of the proposed approach across varying environmental settings. These aspects will be covered in Discussion section of the revised manuscript.

**Other Notes:**
**R: LN 20: Comma needed in "4862"?**
**A**: Will be done

**R: LN 30: May consider highlighting that ANNs have also proven successful for forecasting subsurface hydrologic response for landslide-prone hillslopes. https://doi.org/10.1029/2020GL088731**
**A:** Thanks for indicating the linked study. The following lines may be added in the Introduction Section:
*"Orland et al. (2020) developed a deep learning model that utilized soil moisture, pore pressure, and rainfall monitoring data from landslide-prone hillslopes in the USA to predict both the timing and magnitude of hydrologic responses at various soil depths. Their findings suggest that machine learning offers an accurate and computationally efficient alternative to empirical approaches and physical models for landslide hazard warning. Other studies have also confirmed the effectiveness of machine learning in different contexts, such as Distefano et al. (2022, 2023) [...]"*

**R: LN 34-41: Is it worth mentioning that virtually all of these kinds of rainfall and soil moisture products (e.g., NASA GPM and SMAP) have some kind of latency?**

**A**: We fully agree that latency is a common feature of these kinds of rainfall and soil moisture products. For instance, the NASA GPM (Global Precipitation Measurement) mission provides different data streams with varying latencies between the observation time and the publication varying from 4 hours to 3.5 months (Huffman et al., 2019). Similarly, NASA's SMAP (Soil Moisture Active Passive) soil moisture retrievals typically have a latency of 1–3 days, with latency increasing for higher-level or gap-filled products (Dashtian et al. 2024). Thus, in the revised manuscript, we will highlight that the latency issue affects a broader range of Earth observation datasets, and that the approach presented in this study is not only relevant to ERA5-Land reanalysis dataset, but an issue common to other sources of data that may be of interest for landslide prediction studies.
REFERENCES: Huffman, G.J.; Bolvin, D.; Braithwaite, D.; Hsu, K.; Joyce, R.; Kidd, C.; Nelkin, E.; Sorooshian, S.; Tan, J.; Xie, P. NASA Global Precipitation Measurement (GPM) Integrated Multi-Satellite Retrievals for GPM (IMERG)). Algorithm Theoretical Basis Document (ATBD) Version 06; NASA/GSFC: Greenbelt, MD, USA, 2019; Volume 30
Dashtian H, Young MH, Young BE, McKinney T, Rateb AM, Niyogi D, Kumar S V. (2024) A framework to nowcast soil moisture with NASA SMAP level 4 data using in-situ measurements and deep learning. J Hydrol Reg Stud 56:102020. https://doi.org/10.1016/J.EJRH.2024.102020

**R: LN 57-58: "On the other side," may be unnecessary text.**
**A**: Will be done

**R: LN 133-134: What are the implications for assuming landslide timing as the end of the day?**

**A:** We acknowledge that assuming landslide occurrence at the end of the day introduces temporal uncertainty, which may affect the estimation of rainfall conditions leading to the event. As discussed by Peres et al. (2018), even small shifts in the assumed landslide timing—such as reporting the event earlier or later than it occurred—can significantly affect the derived rainfall thresholds. Nonetheless, it is not possible to overcome this lack of precise timing, and the best we could do is to use the available information consistently throughout all the analyses we have conducted. A comment will be added in the Discussion section of the revised manuscript.

---

## Author Comment (AC2)

**REFEREE'S COMMENTS ON egusphere-2025-1590**

**RC#2**
**GENERAL COMMENT**
**R: The contribution "Use of delayed ERA5-Land soil moisture products for improving landslide early warning" by N. Palazzolo and co-Authors is interesting and addresses questions relevant with the scope of NHESS. In this paper the researchers conducted a series of direct experiments. They aimed to determine how effective it is to use antecedent soil moisture data (ranging from 0 to 15 days old) from a global reanalysis model, combined with rainfall data, to predict landslide triggers using machine learning. As one might expect, the findings indicate that this historical soil moisture data enhances model performance, though its benefit diminishes as the time lag increases. Results, and discussion sections are short compared to amount of work done. They should be increased. The theoretical background is well-argued but not complete. Review of literature seems completed. The description of study area is sufficiently complete. The description of methodology and successive parts of paper are well organized but not complete. The readability of the whole paper is good with a good English. In general, the synthetic approach of the study is very clear, but I think that going into more detail on several aspects (later reported) could be useful for the readers. It can be published on NHESS journal only after minor revision.**

**A:** The authors sincerely thank the reviewer for the detailed review and the insightful comments, which will help us improve the quality of our work. Based on your suggestions, we will expand the methodology section and enriched the results and discussion with additional analyses. Below, we provide detailed point-by-point responses to each comment.

**SPECIFIC COMMENTS**
**R: 1 Introduction (Line 30-40)**
**My main comment regards the conceptual background of the comparing of the performance of different models. It would be necessary to better understand from the authors whether it is possible to directly compare the performance of the models with and without soil moisture (on the empirical rainfall thresholds) information based exclusively on the comparison with the TSS index or whether other strategies exist in this direction.**

**A:** Thank you for the comments. First let us clarify that we compare the performances of ANN models with and without soil moisture, so the comparison of True Skill Statistic (TSS) is made under the same conditions and also has been the main topic of previous papers (Distefano et al., 2022). We acknowledge, however, that relying exclusively on a single performance index such as TSS may not fully capture all aspects of model performance. Although TSS is widely used and recommended in the literature for summarizing model skill in binary classification tasks (Frattini et al, 2010), it also corresponds to the performance closest to the top-left corner of the ROC space, which represents the ideal balance between true positive and false positive rates. For this reason, we selected TSS as the main metric to allow a direct and consistent comparison across models. Nevertheless, we agree that additional analyses can help refine the interpretation. In the revised version of the manuscript, we will therefore include a more comprehensive ROC-based analysis to better characterize the trade-offs between missed alarms and false alarms across the different models. Furthermore, we have adopted strategies for ensuring generalization capabilities of the developed ANNs, by considering training, validation and test datasets and the early stopping technique, for taking into account of the different complexity of models. We will however mention that other metrics such as the Akaike Information Criterion (AIC), Bayesian Information Criterion (BIC), or adjusted performance scores can be useful when model complexity differs.

REFERENCE: P Frattini, G Crosta, A Carrara, Techniques for evaluating the performance of landslide susceptibility models, Engineering Geology, Volume 111, Issues 1–4, 2010, Pages 62-72, ISSN 0013-7952, https://doi.org/10.1016/j.enggeo.2009.12.004.

**2 Material and methods**
**R: (Line 48-49) Have you tried using landslide data from other catalogues? (i.e. Peruccacci et al., 2023)**
**A**: We thank the reviewer for the suggestion. At this stage, we have not yet tested other landslide catalogues. As a first step, we used the same landslide dataset as Distefano et al. (2023) to allow a consistent comparison and to evaluate how our methodology performs under similar conditions. However, we fully agree that testing different landslide inventories (Peruccacci et al., 2023), as well as alternative sources of soil moisture and rainfall data, may be a direction for future

research, in order to strengthen the validity of our findings. Hence, we will add a sentence about this in the Discussion section of the revised manuscript.

Reference: Peruccacci S., Gariano S.L., Melillo M., Solimano M., Guzzetti F., Brunetti M.T. (2023) The ITAlian rainfallinduced LandslIdes CAtalogue, an extensive and accurate spatio-temporal catalogue of rainfall-induced landslides in Italy. Earth Syst Sci Data, 15, 2863–2877, doi: 10.5194/essd-15-2863-2023

**R: (Line 55-60) The Authors reconstruct triggering and non-triggering rainfall events.**
**I didn't understand the following list of things:**
**The rainfall event or landslide triggering condition (MPRC automatically reconstructed by the tool) normally represents a subset of the entire rainfall event while non-triggering rainfall events are always considered as a whole. Did the authors take this difference into account when creating the input data for the neural network?**
**A**: Yes, we confirm that this is the case, even though in practice in most of the cases the entire rainfall events is attributed to the landslide (see next reply).

**R: How many triggering conditions coincide with the entire rainfall event? If not, how many of them differ significantly in terms of duration and cumulative rainfall, and what justifies the choice of these two sets of triggered and non-triggered rainfall events? The authors should provide a better argument and commentary on the method used.**

**A:** For most of the landslide events (103 out of 144) only the day of occurrence was available in the database. In this case we assumed that the triggering instant was at the end of the day. Hence, in many cases the entire rainfall is attributed to landslide triggering. However, we are aware that from a theoretical standpoint it may not be correct to attribute to a landslide the part of the triggering event that goes beyond the triggering event. We may add a comment on this point in the revised manuscript.

**R: The number of reconstructed non-triggering rainfall events is at least two orders of magnitude greater than those that trigger landslides. What procedure was used to account for this substantial difference? Are the samples used for the training, validation and testing phases balanced or unbalanced?**
**A:** Thank you for mentioning this issue. Indeed, the dataset used in this study is unbalanced, and the higher number of non-triggering rainfall events reflects the actual availability of observed landslides in the catalogue. However, this imbalance does not compromise the reliability of the modelling process. ANNs were trained using a cross-entropy loss function, which is widely adopted in classification tasks and known to be robust to class imbalance. Then, the network outputs are transformed into binary predictions through a thresholding procedure, where a landslide is predicted (i.e., output = 1) when the network output exceeds a certain threshold, and 0 otherwise. Within a study currently under preparation we have investigated the influence of different proportions of non-triggering and triggering events. The outcome of these tests suggests that it not necessary to balance the datasets with our model setting and that unbalanced datasets lead to more robust models, as, while using the entire information available, they reflect the actual likelihood of triggering and non-triggering events. A comment on this point will be added to the revised manuscript in the Methodology and Discussion sections.

**R: What observations are used in this last case, and how is sampling performed? Have statistical tests been conducted to confirm whether these samples are representative of the population? Please specify the setting more precisely.**
**A**: We further clarify that the entire dataset, comprising both triggering and non-triggering rainfall events, is randomly split into three subsets: 70% for training, 15% for validation, and 15% for testing. The subset allocation was random, according to a uniform distribution over the dataset indices. This means that each observation had the same probability of being assigned to any of the three subsets, ensuring an unbiased and representative sampling process. To account for the variability introduced by this random sampling, each neural network configuration was trained and evaluated 30 times, each with a different random split. This strategy allows us to assess the robustness of model performance and to simulate realistic operational conditions, where landslide-triggering events are relatively rare. The use of box plots in the results highlights this aspect: the width of the boxes reflects how sensitive the model performance is to the random selection of the subsets. Narrow boxes indicate stable results across different splits, while wider boxes reveal a stronger dependence

on the specific partitioning. Thus, the repeated randomization approach, together with the use of box plots to represent the results, was explicitly adopted to explore and quantify this variability. This point will be stressed in the revised manuscript.

**R: 2.3 Artificial Neural Network models (ANNs)**
**(Line 80-90) ANNs Input variables/data**
**The reconstruction of the rainfall conditions that triggered landslides is performed starting from the assumption that in the place where the landslide occurs the rainfall is the same as that measured on a representative rain gauge (specific criteria dependent on a parameterized variable that considers the distance was used by CTRL-T tool). As regards the choice of cells that contains information associated with the soil moisture status, it seems that the cell that includes the representative rain gauge chosen automatically by the tool has been considered. Under this hypothesis (soil moisture associated with the cell that includes rain gauge) what considerations can be made from a physical point of view? Is it conceptually correct to look for a relationship between the saturation state of a soil different from that in which the triggering of a landslide occurs? If the cell that contains the landslide is considered, do the results change? Considering the spatial resolution of the ERA5-Land product (about 9km x 9km) probably in most cases the cell containing the landslide or the rain gauge is the same, but I believe that deepening this aspect by making descriptive statistics could be useful. Moreover, for an operational use, the forecast value returned as output by the model would describe the possibility that a landslide could happen in that specific cell for which the soil moisture value was provided as input. Therefore, in my opinion, it should perhaps also be used in the model's learning phase.**

**A:** We agree that ideally the rain gauge, soil moisture information and landslide locations should be the same. Nevertheless, this is never possible in practice (with rain gauges) as the location of future landslides is unknown. What our model predicts is "a possible areal landslide event" within a fixed radius $R_b$ around the rain gauge. The performance we have obtained prove that the use of soil moisture, even with this spatial resolution provides better performances respect to the use of precipitation only. This may be related to spatial correlation of soil moisture. A comment will be added on this point in the revised

**R: 3 Study area and data**
**(Line 140-150) CTRL-T parameters setting**
**What is the length of the "warm" spring–summer period (CW) and "cold" autumn–winter period (CC). More in detail, what values were assigned to the variables sws (beginning of warm season) and ews (end of warm season), i.e. CW from May to October and CC from November to April correspond sws=5 (May), ews=10 (October). Please specify the setting more precisely. Add the *sws* and *ews* parameters in the Table 1 (cfr. comment refered to Line 140-150)**
A: Thank you for the opportunity to provide further clarification. In our study, we defined the warm season (CW) as the period from April to October, and the cold season (CC) as the period from November to March. Accordingly, the parameters sws = 4 and ews = 10 will be added to Table 1 for clarity.

**R: 4 Results and discussion**
**(Line 160-165)**
**Figure 4 shows box plots of the results for the autocorrelation conditions. The text provides a good description of the behaviour of the variations in terms of lag k. However, a visual analysis of the (k) values (k = 1, 2, ..., 15 days) in the different layer depth cases (b) 7–28 cm, (c) 28–100 cm, and (d) 100–239 cm reveals that the values are indistinguishable (perhaps due to the scale). Consequently, it is unclear why layer 2 was selected for subsequent analysis (b). Please explain better.**
A: Thank you for this comment. The purpose of Figure 4 is to explore overall the autocorrelation structure of soil moisture by comparing its lagged values (k = 1 to 15 days) with the reference value at lag 0. The box plots are intended to show how the autocorrelation decreases with increasing lag, rather than to directly compare the performance of models obtained from different soil layers' data. As for the choice of Layer 2 (7–28 cm) for subsequent analyses, this was motivated by our previous findings reported in Di Stefano et al. (2023), where layer 2 consistently yielded the best performance in landslide prediction tasks. Based on this, we decided to focus our experiments on the configurations presented within the manuscript. We will further clarify this in our revised manuscript.

**R: Line 175-190**

**How does performance change for all combinations of layer use? For example, what is the maximum TSS (mean) value for D-E-S1, D-E-S3, D-E-S4, D-E-S all-1, and so on? Has a systematic analysis been conducted to better understand the effect of each input data component introduced in network training? If it is not very time-consuming, I would suggest trying this approach.**

**A**: We thank the reviewer for the suggestion. A systematic analysis of the effect of different combinations of input layers (including D, E, S1, S3, S4, and their combinations) on model performance has already been conducted in Distefano et al. (2023), which our work takes as a starting point. For this reason, we did not replicate the same analysis here, as we believe it would not significantly add to the substance of the present study. However, we will better clarify this aspect in our revised manuscript.

**R: Figures description**

**Description of Figure 2**

In the structural diagram of the ANN, the symbol for dichotomy on the third layer is different from those for the three preceding nodes on layer 2.

**A**: We thank the reviewer for the observation. The difference in the symbol is intentional, as two different activation functions have been used: a tan-sigmoid function f(n) for the hidden layer, with output in the range (-1, 1), and a log-sigmoid function g(n) for the output layer, with output in the range (0, 1). Additional details will be added in the revise manuscript to make it clearer.

**R:** I suggest to:

remove "---and---" because it's implicit in the figure

replace "depth" with "cumulated rainfall" (also extended to the text)

*A: Will be done*

**R:** Specify the terms S (ERA5-Land depth)

*A: Will be done*

**R:** Specify which is the input, hidden and output level.

*A: Will be done*

**R: Description of Figure 3**

Add geographical grid and coordinate labels.

*A: Will be done*

**R:** I suggest changing the colour filling of the peninsular part of Italy (i.e. the monochrome palette) because the scale is different respect to the study area.

*A: Will be done*

**R:** I suggest increasing the size of the dots and changing the starting colour of palette from "blue" to "green" for greater contrast.

*A: Will be done*

**R:** Add the description of the symbols in the caption.

*A: Will be done*

**R: Description of Figure 4**

Formatting borders, label text, and ticks in black instead of grey

*A: Will be done*

**R:** Add the description of the continuous red and the dashed grey curves in the caption (also extended to the text).

Remove the labels and headings from the x-axis for (a) and (b) and from the y-axis for (b) and (d). This should make the figure clearer by increasing its size.

*A: Will be done*

**R: Description of Figure 5**

Formatting borders, label text, and ticks in black instead of grey

*A: Will be done*

**R:** Change dimensions of '(a)' and '(b)' as in figure 4.

*A: Will be done*

**TECHNICAL CORRECTIONS**
**R:** Below a list of some more detailed comments and suggestions referred to specific parts of the text.
(Line 54): replace "($D$, in hours)" with "$D$ (h)"
*A: Will be done*
**R:** (Line 55): replace "($E$, in millimeters)" with "$E$ (mm)"
*A: Will be done*
**R:** (Line 116): remove the space in "non-triggering"
*A: Will be done*
**R:** (Line 120): remove the space in "(missed alarms)"
*A: Will be done*
**R:** (Line 123): Change the 'T' term format
*A: Will be done*
**R:** (Line 126): enter the comma in the range [0,1]
*A: Will be done*
**R:** (Line 129): replace "southern Italy" with "Southern Italy"
*A: Will be done*
**R:** (Line 31): replace "700" with "700 mm"
*A: Will be done*

REFERENCE: Peruccacci S., Gariano S.L., Melillo M., Solimano M., Guzzetti F., Brunetti M.T. (2023) The ITAlian rainfall-induced LandslIdes CAtalogue, an extensive and accurate spatio-temporal catalogue of rainfall-induced landslides in Italy. Earth Syst Sci Data, 15, 2863–2877, doi: 10.5194/essd-15-2863-2023

---

## Author Comment (AC3)

REFEREE'S COMMENTS ON egusphere-2025-1590

RC#3

R: This is an interesting study emphasizing the utility of ERA5 soil moisture data for improving landslide forecasting, despite the coarse resolution (9x9km) and considerable latency (5d) of the product. Overall, the results are intuitive and confirm expectations based on previous research, namely that ERA5, which reflects more than just antecedent rainfall, will improve landslide forecasting when paired with rainfall data. This is useful despite these limitations with resolution and latency of ERA5 and I agree with reviewers #1 and #2 that this will ultimately contribute to the literature and be appreciated by readers of NHESS. The explicit evaluation of latency impacts is interesting, though from the perspective of implementation for early warning it's unclear why a hypothetical latency of 15d is useful since the actual product currently has the fixed delays of 5d, particularly when other more practical questions about the broader utility of ERA5 could be investigated (see below).

A: We thank the reviewer for this insightful comment. Although the current operational latency of ERA5-Land soil moisture is approximately 5 days, we chose to explore a broader range of latencies (from 1 to 15 days) to account for both potential future improvements in data delivery and hypothetical scenarios of temporary service disruptions. More broadly, our goal was to assess how forecast performance degrades as data timeliness decreases, and to evaluate the model's sensitivity to delays in soil moisture availability. This analysis helps clarify the operational value of ERA5-Land and identify the latency thresholds within which the product remains useful for early warning purposes.

R: As pointed out by both reviewers, the paper is indeed lacking on discussion and broader implications, so overall, I found the analysis somewhat narrow in terms of the scope of hypotheses tested. As Reviewer #1 noted in his paper for the San Francisco Bay Area, California (Thomas, Collins, and Mirus, WRR, 2019), SMAP data is useful, but in-situ soil moisture sensors have the capacity to improve over the general limitations of satellite soil moisture and rainfall data, even though the latter is theoretically available everywhere on the globe. As we further note in our recent perspective (Mirus, Bogaard, Greco and Stahli, NHESS, 2025), hillslope monitoring stations are advantageous for improving forecasts, but are difficult to maintain and come with other representativeness issues. So, we suggested more rigorous comparison of in-situ sensors from hillslope locations with satellite data for landslide prone areas would shed more light on the utility of satellite data for landslide forecasting. The current study misses this opportunity. Are there any in-situ hydrological data available *anywhere* in your study area to expand the value and impact of your study? I realize that the Contra Costa and Alameda counties from Matt's paper are only ~5,000 km2 whereas Sicily is closer to 26,000 km2, so I'm curious if you'd find over this larger scale that the satellite soil moisture, despite the lags, is still more useful than in-situ sensors for spatially explicit landslide forecasting?
The study is fine otherwise ad comparable to a technical note or methodological contribution. Again, I would urge the authors to dig deeper in their analyses, as all three reviewers suggest, to enhance the impact and utility of this work to inform landslide early warning systems in Italy and worldwide.
Ben

A: We thank the reviewer for this valuable comment and for pointing out the broader discussion around the trade-offs between satellite-derived and in-situ soil moisture data. We fully agree that in-situ hydrological observations, especially from hillslope monitoring stations, are crucial for improving landslide forecasting. However, in our study area there are not publicly available in-situ soil moisture data, nor private data for a sufficient period. The closest point of the International Soil Moisture Network is in a different region, Calabria (https://ismn.earth/en/). Given this fact and considering the regional scale of the analysis (~26,000 km²), we relied on the globally available ERA5-Land reanalysis soil moisture product. While previous studies have shown that ERA5 soil moisture may be less accurate than ground-based observations, especially in complex terrains, our results show that including ERA5 Land data still improves model performance compared to using rainfall alone. This supports the idea that even uncertain soil moisture estimates can provide useful information for landslide forecasting at regional scale. This observation aligns with findings by Marino et al. (2020), who demonstrated that even soil moisture estimates derived from a simplified hydrological model—rather than satellite or in-situ data—can significantly enhance landslide prediction. In their study, the authors used a Monte Carlo approach to assess the robustness of forecasting thresholds under different uncertainty scenarios, and they found that soil moisture information consistently contributed to reducing false alarms. These results highlight that the added value of

soil moisture information is not confined to localized, high-accuracy observations, but may also emerge from coarser, large-scale products that adequately capture the temporal patterns relevant to slope instability. We will include these considerations in the revised discussion section. We also agree that our manuscript was quite short in some parts. With the revisions following referee comments we believe that we will be able to expand it to make it less comparable to a technical note.

---

## Author Comment (AC4)

REFEREE'S COMMENTS ON egusphere-2025-1590

RC#4
**R: The paper is presented in the format of a short communication or technical note aimed at providing insights into the added value associated with the use of volumetric soil water content as provided by ERA5-Land, as an additional proxy for event identification. I agree with the points raised by the other reviewers: I appreciate the clarity and simplicity of the approach. All steps are straightforward and easy to understand; nonetheless, I also share some concerns (and it would therefore be useful to provide further information regarding):**

**A**: Thank you for your comment. While we understand that the structure of our manuscript may resemble a short communication, it does not meet the criteria defined by NHESS for that format, which is limited to 2–4 journal pages. Based on the constructive feedback received from all four reviewers, we are currently expanding the manuscript and integrating additional analyses to enhance its scientific depth and clarity.

**R: What is the current early warning system used in Sicily for landslide forecasting, and what is the added value compared to this benchmark? (In this regard, if the system is based solely on triggering rainfall, it would be very interesting to understand the added value of using soil moisture as an event discriminator);**

*A*: Currently there is no early warning system in Sicily specific for landslides. There is indeed an alert system for generic "hydro-geological risk" which is generic for floods and landslides and is based on comparing quantitative precipitation forecasts (QPF) with precipitation depth-duration-frequency curves of different rainfall return periods. Hence, it is based on intrinsic characteristics of rainfall, which is analyzed independently from the observed impacts produced by extreme rainfall. For instance, an alert level is activated when rainfall is predicted to exceed the IDF curve of 0.80 of non-exceedance probability, and this is not necessarily linked to observed floods/landslides in an area, as this depends on catchment/soil characteristics which are not taken into account by the system. Indeed, several theoretical inconsistencies of this approach have been highlighted in a recent brief communication under review in NHESS (Marra et al., 2025). Given this, inclusion of soil moisture and the increase of performance respect to precipitation-based thresholds shown in our analyses already prove the added value of using soil moisture. We will add a sentence to the discussion section of our revised manuscript.

REFERENCE: Marra F, Dallan E, Borga M, Greco R, Bogaard T Brief communication: Threshold not probability. The conceptual difference between ID thresholds for landslide initiation and IDF curves. https://doi.org/10.5194/egusphere-2025-3378

**R: what are the characteristics of the movements in the area? It is obviously well known that the added value of using volumetric water content as a proxy depends on the characteristics of the event—events in finer-grained soils are more influenced by antecedent rainfall, while with thinner soil layers or higher permeability, the importance of triggering precipitation increases. With that in mind, is it possible to identify a sort of zonation to understand in which areas there might be added value in using reanalysis-based volumetric soil water content as a proxy? If possible, a spatial representation of the information could be very helpful.**

**A**: We agree with the reviewer that the characteristics of landslide movements can strongly influence the added value of using soil moisture information. In principle, it would be meaningful to perform separate analyses based on the type of movement or according to the geographical alert zones used in Sicily's civil protection system. However, as also noted in our response to Referee #1, the type of movement is unfortunately unknown for most of the 144 landslide events included in our dataset. Additionally, splitting the dataset by the 9 official alert zones would result in very small subsets, which would not be sufficient to support a robust statistical evaluation of model performance. For these reasons, we chose not to apply a spatial or typological stratification in the present analysis. Nonetheless, we acknowledge the potential value of this approach and consider it a promising direction for future work, particularly as more detailed and spatially distributed landslide information becomes available. A comment on this will be added to the revised manuscript.

**R: while I understand the rationale for using information with up to a 15-day delay in the first part of the analysis, from the introduction of the ANN approaches onward, I find it less useful and would limit all representations to 5**

**days before the present time; it is likely that in upcoming releases (ERA6 is expected in December 2025 with 14 km resolution) the delay will decrease and not increase.**

A: We understand the reviewer's concern regarding the limited operational relevance of testing latencies beyond 5 days, particularly in the context of ANN-based forecasting. However, we believe that exploring a wider latency window (up to 15 days) remains meaningful for two main reasons. First, while the current latency of ERA5-Land is about 5 days and future products such as ERA6 may further reduce this delay, temporary disruptions (e.g., due to maintenance or data access issues) cannot be excluded in operational contexts. Second, from a scientific standpoint, our aim was to investigate the sensitivity of model performance to soil moisture availability over time. This allows us to better understand the degradation of performance as information becomes less timely. We agree that, in practical terms, the most relevant scenarios are within the 1–5 days window, and this is reflected in our discussion of operational perspectives. We appreciate the reviewer's suggestion and will make this clearer in the revised manuscript.

**R: although it is well known to most, for completeness I would include a brief description of the ERA5-Land reanalysis (it would also be helpful to emphasize that 9 km is the resolution of the land module only, while the atmospheric part is a statistical interpolation from the parent model "ERA5"). For accuracy, I would avoid referring to ERA5-Land as "products" or "dataset" and instead use the term "reanalysis outputs."**

*A:* We thank the reviewer for the helpful suggestion. In the revised manuscript, we will include a brief description of the ERA5-Land reanalysis to improve completeness and clarity and to emphasize its relationship with ERA5. Thanks for your suggestion about the use of correct terminology. However, we have chosen to use the terms "dataset" in accordance with the official terminology adopted by the Copernicus Climate Data Store (CDS), which distributes ERA5-Land (see https://cds.climate.copernicus.eu/datasets/reanalysis-era5-land?tab=overview). Also "reanalysis product" is consistent with the data provider and broader scientific usage (see https://link.springer.com/article/10.1007/s00382-023-06803-w). Thus, we prefer to maintain this terminology in our manuscript.

**R: Additionally, I note: In the abstract, avoid or explain any acronyms used.**
*A: Will be done*
**R: there's a typo in Figure 1.**
*A: Will be fixed*

---

## Author Response (AR1)

REFEREE'S COMMENTS ON egusphere-2025-1590

RC#1
**R: Thank you for the opportunity to review this short-form manuscript. The authors design a set of straightforward experiments that include testing the efficacy of using 0- to 15-day antecedent soil moisture information from a modeled global reanalysis data product, in conjunction with rainfall data, to identify the triggering conditions for landslides using machine learning. The Results and Conclusion are intuitive in that antecedent soil moisture improves model performance, with the benefit decreasing somewhat with increased lag. This deprecation in model performance seems minor for a lag that is equivalent to the latency of the modeled soil moisture data product (~5 days). Although I appreciate the streamlined presentation of this study, I think it would be helpful for readers to see more text related to (1) the kind of landslides this study is relevant to, (2) why the spatiotemporal resolution of the modeled soil moisture data product is appropriate for the landslide type(s) considered here, and (3) a deeper interpretation of the Results. Regarding #3 - What are the rainfall depth/duration characteristics and the antecedent soil moisture levels that the best-performing model settles on? And do these characteristics make sense relative to the landslide type(s) and/or any previously published regional thresholds? The objective of this study is crystal clear, but the authors may consider questions like these to expand the relevance of their work for the broader scientific community.**
**Sincerely,**
**Matthew A. Thomas**

A: Thank you for your general appreciation of our manuscript and your valuable comments. In the following, we reply by showing the manuscript revisions applied:

1) Regarding the type of landslides our study is relevant to, as well as the spatiotemporal resolution of the modeled soil moisture, the following lines have been added to our revised manuscript within the "Study area and data" section:
"*Furthermore, the mentioned landslide database distinguishes between two categories of events: single landslide events (SLEs) and areal landslide events (ALEs). SLEs generally provide more precise temporal and spatial information on the failure, whereas ALEs consist of multiple landslides within a defined area and are typically associated with lower spatial resolution—often limited to administrative units such as municipalities. Despite the potential uncertainties in the location and data for both SLEs and ALEs, we have chosen to retain all events in order to have statistically representative subsamples for training, validating and testing the ANNs. More specifically, only those landslides linked to rainfall as the main triggering factor were considered. This includes events where the mobilized material was consistent with rainfall-induced landslides, while other types (e.g., rockfalls) were excluded. The final dataset consisted of 207 rainfall-related landslides. The CTRL-T software enabled the reconstruction of triggering conditions for 144 of these events. Among them, movement type information is unavailable for 126 events (87.5%). Of the remaining cases, 10 were identified as areal rockfalls (6.9%), and both flows and slides accounted for 4 events each (5.6%). Regarding temporal resolution, for 103 landslides only the day of occurrence was known, while for the remaining cases more detailed information— such as the hour or time of the day (e.g., morning, afternoon, or evening)—was available. For events with daily resolution, the failure was assumed to occur at the end of the day. When more precise timing was reported, the failure was assumed to coincide with the peak rainfall within the reported time window. However, as discussed by Peres et al. (2018), even small shifts in the assumed timing of landslide occurrence—whether anticipating or delaying the event—can affect the estimation of rainfall thresholds. Despite this limitation, the lack of precise temporal information cannot currently be resolved. Nonetheless, the consistency of the dataset across all models ensures an objective comparison of their performance.*"

2) Regarding a deeper interpretation of the results, i.e. more details on the rainfall depth/duration characteristics and the antecedent soil moisture levels that the best-performing model settles on, and whether these characteristics make sense relative to the landslide type(s) and/or any previously published regional thresholds, "Results" section have been enriched with the following lines:
"*Concerning the generalizability and broader applicability of the proposed approach, it is worth noting that the rainfall events considered for model development covered a wide range of characteristics, with durations*

*between 1 hour and 10 days and cumulative totals up to 300 mm (Distefano, 2023). In addition, soil moisture values ranged between 0.15 and 0.45, effectively spanning the porosity domain of most common soil types. These conditions delineate the applicability domain of the trained models and support their potential transferability to different climate and geological conditions".*

**Other Notes:**
**R: LN 20: Comma needed in "4862"?**
**A**: Done

**R: LN 30: May consider highlighting that ANNs have also proven successful for forecasting subsurface hydrologic response for landslide-prone hillslopes. https://doi.org/10.1029/2020GL088731**
**A:** Thanks for indicating the linked study. The following lines have been added in the Introduction Section:
*"[…] Meanwhile, other studies have explored the benefit from the incorporation of soil-hydrologic monitoring data into ML-based models to simulate changes in soil moisture and pore pressure that predispose triggering mechanisms. Orland et al., (2020), for instance, developed a deep learning model that combined soil moisture, pore pressure, and rainfall monitoring data from landslide-prone hillslopes in the USA to predict both the timing and magnitude of hydrologic responses at various soil depths. Their findings suggest that machine learning can provide an accurate and computationally efficient alternative to empirical approaches and physical models for landslide hazard warning."*

**R: LN 34-41: Is it worth mentioning that virtually all of these kinds of rainfall and soil moisture products (e.g., NASA GPM and SMAP) have some kind of latency?**

**A**: We fully agree that latency is a common feature of these kinds of rainfall and soil moisture products. Thus, we added the following lines within the "Introduction" section:
*"Indeed, such a latency issues affects a broader range of Earth observation datasets. For instance, the NASA GPM (Global Precipitation Measurement) mission provides datasets with varying latencies, between the observation time and the publication one, varying from 4 hours to 3.5 months (Huffman et al., 2019). Similarly, NASA's SMAP (Soil Moisture Active Passive) soil moisture retrievals typically have a latency of 1–3 days, with latency increasing for higher-level or gap-filled products (Dashtian et al., 2024). Against this backdrop, the present study investigates the feasibility of using specifically delayed ERA5-Land soil moisture data for real-time landslide forecasting, recognizing that data latency is a common challenge shared by other sources potentially valuable for landslide prediction studies."*

REFERENCES: Huffman, G.J.; Bolvin, D.; Braithwaite, D.; Hsu, K.; Joyce, R.; Kidd, C.; Nelkin, E.; Sorooshian, S.; Tan, J.; Xie, P. NASA Global Precipitation Measurement (GPM) Integrated Multi-Satellite Retrievals for GPM (IMERG)). Algorithm Theoretical Basis Document (ATBD) Version 06; NASA/GSFC: Greenbelt, MD, USA, 2019; Volume 30
Dashtian, H., Young, M. H., Young, B. E., McKinney, T., Rateb, A. M., Niyogi, D., and Kumar, S. V.: A framework to nowcast soil moisture with NASA SMAP level 4 data using in-situ measurements and deep learning, J. Hydrol. Reg. Stud., 56, 102020, https://doi.org/10.1016/J.EJRH.2024.102020, 2024.

**R: LN 57-58: "On the other side," may be unnecessary text.**
**A**: Done

**R: LN 133-134: What are the implications for assuming landslide timing as the end of the day?**
**A:** Please, see reply to point 1).

**RC#2**
**GENERAL COMMENT**
**R: The contribution "Use of delayed ERA5-Land soil moisture products for improving landslide early warning" by N. Palazzolo and co-Authors is interesting and addresses questions relevant with the scope of NHESS. In this paper the researchers conducted a series of direct experiments. They aimed to determine how effective it is to use antecedent soil moisture data (ranging from 0 to 15 days old) from a global reanalysis model, combined with rainfall data, to predict landslide triggers using machine learning. As one might expect, the findings indicate that**

this historical soil moisture data enhances model performance, though its benefit diminishes as the time lag increases. Results, and discussion sections are short compared to amount of work done. They should be increased. The theoretical background is well-argued but not complete. Review of literature seems completed. The description of study area is sufficiently complete. The description of methodology and successive parts of paper are well organized but not complete. The readability of the whole paper is good with a good English. In general, the synthetic approach of the study is very clear, but I think that going into more detail on several aspects (later reported) could be useful for the readers. It can be published on NHESS journal only after minor revision.

**A:** The authors sincerely thank the reviewer for the detailed review and the insightful comments. Based on your suggestions, we expanded the methodology section and enriched the result's discussion. Below, we highlight the manuscript changes applied.

**SPECIFIC COMMENTS**
**R: 1 Introduction (Line 30-40)**
**My main comment regards the conceptual background of the comparing of the performance of different models. It would be necessary to better understand from the authors whether it is possible to directly compare the performance of the models with and without soil moisture (on the empirical rainfall thresholds) information based exclusively on the comparison with the TSS index or whether other strategies exist in this direction.**

**A:** Thank you for this comment. We have added some more details, within the "Artificial Neural Network models (ANNs)" section, about ways to compare thresholds as follows:

*"Thus, the network outputs, originally ranging from 0 to 1, were converted into binary predictions through a thresholding procedure, whereby an event is classified as a landslide (i.e., output = 1) if the model output exceeds a predefined threshold, and as a non-triggering event otherwise. This threshold was selected by maximizing the TSS, which ensures an optimal trade-off between TPR and FPR. While TSS is widely adopted in the literature for evaluating binary classifiers (Frattini et al., 2010), other metrics can also be employed to assess model performance and complexity. For example, information criteria such as the Akaike Information Criterion (AIC) and the Bayesian Information Criterion (BIC) have been effectively applied in studies comparing heterogeneous or hybrid models (Dutta et al., 2025; Patton et al., 2023; Quraishi and Choudhury, 2023) to balance model fit and parametrization. In the case of neural networks, however, the most common strategies rely on the use of independent training, validation, and test datasets, combined with early stopping, which—as in the present study—effectively control overfitting and ensure generalization."*

Furthermore, the discussion of the results has been enriched trough an insight in terms of TPR and FPR, as follows:

*"More details about simulations carried out are summarized in Tables 3 and 4. For the E-D-S2 configuration (Table 3), short delays up to 5 days determine a modest reduction in TSS, with TPR almost unchanged (TPR from 0.89 to 0.87) and a moderate increase in the false alarms rate (FPR from 0.12 to 0.16). At longer delays, performance stabilized around TSS = 0.68–0.70, still with relatively high TPR but with higher false alarm rates (up to 0.20). When soil moisture from all four layers was included (Table 4), performances improve slightly systematically; but for instance, with lags up to 5 days, TPR is maintained (TPR = 0.89–0.90), and FPR increases only slightly (up to 0.18). Beyond 5 days, performance gradually declines (TSS = 0.67–0.69, FPR ≈ 0.20–0.21). Compared to the single-layer setup, the multilayer configuration consistently yielded higher TSS values and lower FPRs, confirming that the vertical soil moisture profile provides a more realistic representation of antecedent hydrological conditions. Nevertheless, the increase in performance respect to the most representative layer may not be worth the increase of input complexity."*

REFERENCE: Frattini, P., Crosta, G., and Carrara, A.: Techniques for evaluating the performance of landslide susceptibility models, Eng. Geol., 111, 62–72, https://doi.org/10.1016/J.ENGGEO.2009.12.004, 2010.
Dutta, K., Poddar, A., Middya, A. I., and Roy, S.: Spatial variations of landslide severity with respect to meteorological and soil related factors, Nat. Hazards, 121, 3267–3291, https://doi.org/10.1007/S11069-024-06930-5/FIGURES/5, 2025.

Patton, A. I., Luna, L. V., Roering, J. J., Jacobs, A., Korup, O., and Mirus, B. B.: Landslide initiation thresholds in data-sparse regions: Application to landslide early warning criteria in Sitka, Alaska, USA, Nat. Hazards Earth Syst. Sci., 23, 3261–3284, https://doi.org/10.5194/NHESS-23-3261-2023, 2023.

Quraishi, M. I. and Choudhury, J. P.: Assessment, Categorisation and Prediction of the Landslide-Affected Regions Using Soft Computing and Clustering Techniques, J. Inst. Eng. Ser. B, 104, 579–602, https://doi.org/10.1007/S40031-023-00876-1/TABLES/1, 2023.

**2 Material and methods**

**R: (Line 48-49) Have you tried using landslide data from other catalogues? (i.e. Peruccacci et al., 2023)**

A: We thank the reviewer for the suggestion. At this stage, we have not yet tested other landslide catalogues. As a first step, we used the same landslide dataset as Distefano et al. (2023) to allow a consistent comparison and to evaluate how our methodology performs under similar conditions. This aspect has been clarified within the "Overview and dataset creation" section:

*"Regarding landslide information, in the present study, solely the FraneItalia inventory has been employed, in order to objectively compare the performance of the models using delayed soil moisture with our previous study utilizing non-delayed soil moisture (Distefano et al., 2023). Nonetheless, the proposed approach is fully transferable to other landslide datasets, such as ITALICA (ITAlian rainfall-induced LandslIdes CAtalogue) (Peruccacci et al., 2023), as far as the necessary information is available."*

REFERENCE: Peruccacci S., Gariano S.L., Melillo M., Solimano M., Guzzetti F., Brunetti M.T. (2023) The ITAlian rainfallinduced LandslIdes CAtalogue, an extensive and accurate spatio-temporal catalogue of rainfall-induced landslides in Italy. Earth Syst Sci Data, 15, 2863–2877, doi: 10.5194/essd-15-2863-2023

**R: (Line 55-60) The Authors reconstruct triggering and non-triggering rainfall events.**
**I didn't understand the following list of things:**
**The rainfall event or landslide triggering condition (MPRC automatically reconstructed by the tool) normally represents a subset of the entire rainfall event while non-triggering rainfall events are always considered as a whole. Did the authors take this difference into account when creating the input data for the neural network? How many triggering conditions coincide with the entire rainfall event? If not, how many of them differ significantly in terms of duration and cumulative rainfall, and what justifies the choice of these two sets of triggered and non-triggered rainfall events? The authors should provide a better argument and commentary on the method used.**

A: We confirm that the tool in general produces triggering events that are shorter than the entire rainfall events. The ANN models are used taking as input the rainfall events reconstructed by the CTRL-T tool as they are.

**R: The number of reconstructed non-triggering rainfall events is at least two orders of magnitude greater than those that trigger landslides. What procedure was used to account for this substantial difference? Are the samples used for the training, validation and testing phases balanced or unbalanced? What observations are used in this last case, and how is sampling performed? Have statistical tests been conducted to confirm whether these samples are representative of the population? Please specify the setting more precisely.**

A: Thank you for mentioning this issue. Indeed, the dataset used in this study is unbalanced, and the higher number of non-triggering rainfall events reflects the actual availability of observed landslides in the catalogue. However, this imbalance does not compromise the reliability of the modelling process. Thus, to better clarify this aspect, the following lines have been added:

*"The ANN models were developed using the MATLAB® Deep Learning Toolbox. Training was performed with the scaled conjugate gradient backpropagation algorithm (Møller, 1993), and model performance was evaluated using the cross-entropy loss function (Kline and Berardi, 2005), which is widely adopted in classification tasks and is known to be robust to class imbalance. This ensures a reliable training process even under unbalanced conditions, where the number of reconstructed non-triggering rainfall events could exceed that of triggering events by multiple orders of magnitude. Indeed, within a study currently under preparation, we have investigated the influence of different proportions of non-triggering and triggering events. The outcome of these tests suggests that it not necessary to balance the datasets with our model setting and that even unbalanced datasets lead to more robust models, as, while using the entire information available, they reflect the actual likelihood of triggering and non-triggering events. Hence, the entire dataset, comprising*

*both triggering and non-triggering rainfall events, was randomly split into three subsets: 70% for training, 15% for validation, and 15% for testing. The allocation was performed randomly according to a uniform distribution over the dataset indices, meaning that each observation had the same probability of being assigned to any of the three subsets. To account for the variability introduced by this random sampling process, each neural network configuration was trained and evaluated 30 times, each with a different random split. This approach allows for a robust assessment of model performance and simulates realistic operational conditions, where landslide-triggering events are relatively rare"*

The following lines have also been added to the "Results" section:

*"Overall, box-plots present a narrow interquartile range, supporting the adequacy of the input dataset and confirming robustness of the results. Their width further depicts how sensitive the model performance is to the random selection of subsets, with narrow boxes indicating stable outcomes across different splits.".*

**R: 2.3 Artificial Neural Network models (ANNs)**
**(Line 80-90) ANNs Input variables/data**
**The reconstruction of the rainfall conditions that triggered landslides is performed starting from the assumption that in the place where the landslide occurs the rainfall is the same as that measured on a representative rain gauge (specific criteria dependent on a parameterized variable that considers the distance was used by CTRL-T tool). As regards the choice of cells that contains information associated with the soil moisture status, it seems that the cell that includes the representative rain gauge chosen automatically by the tool has been considered. Under this hypothesis (soil moisture associated with the cell that includes rain gauge) what considerations can be made from a physical point of view? Is it conceptually correct to look for a relationship between the saturation state of a soil different from that in which the triggering of a landslide occurs? If the cell that contains the landslide is considered, do the results change? Considering the spatial resolution of the ERA5-Land product (about 9km x 9km) probably in most cases the cell containing the landslide or the rain gauge is the same, but I believe that deepening this aspect by making descriptive statistics could be useful. Moreover, for an operational use, the forecast value returned as output by the model would describe the possibility that a landslide could happen in that specific cell for which the soil moisture value was provided as input. Therefore, in my opinion, it should perhaps also be used in the model's learning phase.**

**A:** We agree that ideally the rain gauge, soil moisture information and landslide locations should be the same. Nevertheless, this is never possible in practice (with rain gauges) as the location of landslides to predict in the future is unknown. What our model predicts is "a possible areal landslide event" within a fixed radius $R_b$ around the rain gauge. The following lines have thus been added within "Study area and data" section to better clarify this aspect:

*"[...] Specifically, the $R_b$ value of 16 km was used; however, since the mean distance between rain gauges and landslides was approximately 5 km, this maximum value was seldom reached."*

**R: 3 Study area and data**
**(Line 140-150) CTRL-T parameters setting**
**What is the length of the "warm" spring–summer period (CW) and "cold" autumn–winter period (CC). More in detail, what values were assigned to the variables sws (beginning of warm season) and ews (end of warm season), i.e. CW from May to October and CC from November to April correspond sws=5 (May), ews=10 (October). Please specify the setting more precisely. Add the *sws* and *ews* parameters in the Table 1 (cfr. comment refered to Line 140-150)**
**A**: Thank you for the opportunity to provide further clarification. The following details have been revised/added within "Study area and data" section:

*"The warm season CW was defined as the period from April to October, and the cold season CC as the period from November to March. Accordingly, parameters sws and ews, indicating the beginning of the warm season the end of the warm season, are set equal to 4 (i.e., April) and 10 (i.e., October), respectively. These values are consistent with those proposed by Melillo et al., (2015)."*

*Table 1: CTRL-T parameters for the reconstruction of the rainfall events used in the present study (after Distefano et al., 2022).*

| $G_S$ (mm) | $E_R$ (mm) | $R_B$ (km) | *sws* (-) | *ews* (-) | $P_1$ (h) | | $P_2$ (h) | | $P_3$ (h) | | $P_4$ (h) | |
|---|---|---|---|---|---|---|---|---|---|---|---|---|
| | | | | | $C_w$ | $C_c$ | $C_w$ | $C_c$ | $C_w$ | $C_c$ | $C_w$ | $C_c$ |
| 0.2 | 0.2 | 16 | 4 | 10 | 3 | 6 | 6 | 12 | 1 | 1 | 48 | 96 |

**R: 4 Results and discussion**

**(Line 160-165)**

**Figure 4 shows box plots of the results for the autocorrelation conditions. The text provides a good description of the behaviour of the variations in terms of lag k. However, a visual analysis of the (k) values (k = 1, 2, ..., 15 days) in the different layer depth cases (b) 7–28 cm, (c) 28–100 cm, and (d) 100–239 cm reveals that the values are indistinguishable (perhaps due to the scale). Consequently, it is unclear why layer 2 was selected for subsequent analysis (b). Please explain better. How does performance change for all combinations of layer use? For example, what is the maximum TSS (mean) value for D-E-S1, D-E-S3, D-E-S4, D-E-S all-1, and so on? Has a systematic analysis been conducted to better understand the effect of each input data component introduced in network training? If it is not very time-consuming, I would suggest trying this approach.**

**A**: Thank you for this comment. The purpose of Figure 4 is to explore overall the autocorrelation structure of soil moisture by comparing its lagged values (k = 1 to 15 days) with the reference value at lag 0. The box plots are intended to show how the autocorrelation decreases with increasing lag, rather than to directly compare the performance of models obtained from different soil layers' data. Furthermore, a systematic analysis of the effect of different combinations of input layers (including D, E, S1, S3, S4, and their combinations) on model performance has already been conducted in Distefano et al. (2023), which our work takes as a starting point. For this reason, we did not replicate the same analysis here, as we believe it would not significantly add to the substance of the present study. The following lines have been added as comment to Fig.4 to better clarify this aspect in our revised manuscript.

*"[...] For subsequent analyses, as outlined in the previous sections, we focused on layer 2 (7–28 cm), which had already been identified in Di Stefano et al. (2023). This because, in the same study, a systematic evaluation of alternative input combinations (D, E, S₁, S₂, S₃, S₄, and their subsets) indicates that layer 2 consistently provided the best model performance."*

**R: Figures description**

**Description of Figure 2**

In the structural diagram of the ANN, the symbol for dichotomy on the third layer is different from those for the three preceding nodes on layer 2.

**A**: We thank the reviewer for the observation. The difference in the symbol is intentional, as two different activation functions have been used: a tan-sigmoid function f(n) for the hidden layer, with output in the range (-1, 1), and a log-sigmoid function g(n) for the output layer, with output in the range (0, 1).

**R:** I suggest to:

remove "---and---" because it's implicit in the figure

replace "depth" with "cumulated rainfall" (also extended to the text)

*A: Done*

**R:** Specify the terms S (ERA5-Land depth)

*A: Done*

**R:** Specify which is the input, hidden and output level.

*A: Done*

**R: Description of Figure 3**

Add geographical grid and coordinate labels.

*A: Done*

**R:** I suggest changing the colour filling of the peninsular part of Italy (i.e. the monochrome palette) because the scale is different respect to the study area.

*A: Done*

**R:** I suggest increasing the size of the dots and changing the starting colour of palette from "blue" to "green" for greater contrast.

*A: Done*

**R:** Add the description of the symbols in the caption.

*A: Done*

**R: Description of Figure 4**

Formatting borders, label text, and ticks in black instead of grey

*A: Done*

**R:** Add the description of the continuous red and the dashed grey curves in the caption (also extended to the text). Remove the labels and headings from the x-axis for (a) and (b) and from the y-axis for (b) and (d). This should make the figure clearer by increasing its size.

*A: Done*

**R: Description of Figure 5**

Formatting borders, label text, and ticks in black instead of grey

*A: Done*

**R:** Change dimensions of '(a)' and '(b)' as in figure 4.

*A: Done*

**TECHNICAL CORRECTIONS**

**R:** Below a list of some more detailed comments and suggestions referred to specific parts of the text.

(Line 54): replace "($D$, in hours)" with "$D$ (h)"

*A: Done*

**R:** (Line 55): replace "($E$, in millimeters)" with "$E$ (mm)"

*A: Done*

**R:** (Line 116): remove the space in "non-triggering"

*A: Done*

**R:** (Line 120): remove the space in "(missed alarms)"

*A: Done*

**R:** (Line 123): Change the 'T' term format

*A: Done*

**R:** (Line 126): enter the comma in the range [0,1]

*A: Done*

**R:** (Line 129): replace "southern Italy" with "Southern Italy"

*A: Done*

**R:** (Line 31): replace "700" with "700 mm"

*A: Done*

RC#3

**R: This is an interesting study emphasizing the utility of ERA5 soil moisture data for improving landslide forecasting, despite the coarse resolution (9x9km) and considerable latency (5d) of the product. Overall, the results are intuitive and confirm expectations based on previous research, namely that ERA5, which reflects more than just antecedent rainfall, will improve landslide forecasting when paired with rainfall data. This is useful despite these limitations with resolution and latency of ERA5 and I agree with reviewers #1 and #2 that this will ultimately contribute to the literature and be appreciated by readers of NHESS. The explicit evaluation of latency impacts is interesting, though from the perspective of implementation for early warning it's unclear why a hypothetical latency of 15d is useful since the actual product currently has the fixed delays of 5d, particularly when other more practical questions about the broader utility of ERA5 could be investigated (see below).**

**A**: We thank the reviewer for this insightful comment. Although the current operational latency of ERA5-Land soil moisture is approximately 5 days, we chose to explore a broader range of latencies (from 1 to 15 days) to account for both potential future improvements in data delivery and hypothetical scenarios of temporary service disruptions. More broadly,

our goal was to assess how forecast performance degrades as data timeliness decreases, and to evaluate the model's sensitivity to delays in soil moisture availability. This analysis helps clarify the operational value of ERA5-Land and identify the latency thresholds within which the product remains useful for early warning purposes. Thus, the following lines have been added within the "Introduction" section:

*"[...] Against this backdrop, the present study investigates the feasibility of using specifically delayed ERA5-Land soil moisture data for real-time landslide forecasting, recognizing that data latency is a common challenge shared by other sources potentially valuable for landslide prediction studies. Within this framework, we specifically assess the extent to which different publication lags (up to 15 days) affect the performance of ANN-based landslide prediction. Indeed, although the current latency of ERA5-Land soil moisture data is approximately 5 days, this study explores a broader latency range to evaluate the sensitivity of model performance to delayed information. This choice serves a dual purpose: first, to account for potential improvements in future reanalysis products, e.g., ERA6 (https://climate.copernicus.eu/sites/default/files/2022-09/S3_Hans_Hersbach_v1.pdf) as well as temporary disruptions in data availability due to maintenance or access issues; second, to investigate how the timeliness of soil moisture data affects landslide prediction capability. Understanding the extent to which performance decreases as latency increases allows us to better define the operational value of reanalysis data and identify time thresholds within which delayed soil moisture information remains effective for early warning applications."*

**R: As pointed out by both reviewers, the paper is indeed lacking on discussion and broader implications, so overall, I found the analysis somewhat narrow in terms of the scope of hypotheses tested. As Reviewer #1 noted in his paper for the San Francisco Bay Area, California (Thomas, Collins, and Mirus, WRR, 2019), SMAP data is useful, but in-situ soil moisture sensors have the capacity to improve over the general limitations of satellite soil moisture and rainfall data, even though the latter is theoretically available everywhere on the globe. As we further note in our recent perspective (Mirus, Bogaard, Greco and Stahli, NHESS, 2025), hillslope monitoring stations are advantageous for improving forecasts, but are difficult to maintain and come with other representativeness issues. So, we suggested more rigorous comparison of in-situ sensors from hillslope locations with satellite data for landslide prone areas would shed more light on the utility of satellite data for landslide forecasting. The current study misses this opportunity. Are there any in-situ hydrological data available *anywhere* in your study area to expand the value and impact of your study? I realize that the Contra Costa and Alameda counties from Matt's paper are only ~5,000 km2 whereas Sicily is closer to 26,000 km2, so I'm curious if you'd find over this larger scale that the satellite soil moisture, despite the lags, is still more useful than in-situ sensors for spatially explicit landslide forecasting? The study is fine otherwise ad comparable to a technical note or methodological contribution. Again, I would urge the authors to dig deeper in their analyses, as all three reviewers suggest, to enhance the impact and utility of this work to inform landslide early warning systems in Italy and worldwide.**
**Ben**

A: We thank the reviewer for this valuable comment and for pointing out the broader discussion around the trade-offs between satellite-derived and in-situ soil moisture data. We agree that in-situ hydrological observations, especially from hillslope monitoring stations, have the capacity to improve landslides prediction models, over the general limitations of satellite or reanalysis soil moisture data. However, in our study area there are not publicly available in-situ soil moisture data, nor private data for a sufficient period that could allow us to deep the study in this direction. For reference, even the International Soil Moisture Network (https://ismn.earth/en/) show that the closest point to our study area is located within a different region, namely Calabria. Hence, given this limitation affecting our study area as well as the regional scale of the analysis (~26,000 km²), at the present stage, one can only rely on indirect soil moisture data reconstructions, such as the ERA5-Land reanalysis soil moisture. Our results show that including ERA5-Land data still improves model performance compared to using rainfall alone. This supports the idea that even uncertain soil moisture estimates can provide useful information for landslide forecasting at the regional scale. These findings are in line with those by Marino et al. (2020), where they have demonstrated that even soil moisture estimates derived from a simplified hydrological model—rather than satellite or in-situ data—can significantly enhance the robustness of forecasting thresholds under different uncertainty scenarios by reducing false alarms. These results highlight that the added value of soil moisture information is not confined to localized, high-accuracy observations, but may also emerge from coarser, large-scale products that adequately capture the temporal patterns relevant to slope instability. Thus, we included these considerations in the revised discussion section, as follows:

*"Our results corroborate the findings of Marino et al. (2020), where, based on Monte Carlo simulations, they have shown that soil moisture estimates, even when derived from a simplified hydrological model—rather than satellite or in-situ data—can significantly enhance the robustness of forecasting thresholds under different uncertainty scenarios by reducing false alarms. These findings emphasize that high-accuracy, point-scale observations, which have proven to have more information content than satellite data (Mirus et al., 2025), are not strictly necessary to extract meaningful predictive signals. On the contrary, large-scale products, when properly interpreted, can effectively reflect the temporal dynamics of soil moisture that control slope stability. Therefore, satellite or reanalysis-based soil moisture datasets represent a valuable resource to support landslide early warning systems, especially in areas lacking soil moisture monitoring networks, as in Sicily, where there are no publicly available datasets – for instance, the closest sensors from the International Soil Moisture Network ([https://ismn.earth/en/](https://ismn.earth/en/)) are in Calabria. This is particularly relevant in contexts where current warning procedures rely solely on rainfall thresholds, as the inclusion of soil moisture, even coarse or delayed, provides useful information on catchment predisposition and can improve the reliability of alerts."*

**RC#4**

**R: The paper is presented in the format of a short communication or technical note aimed at providing insights into the added value associated with the use of volumetric soil water content as provided by ERA5-Land, as an additional proxy for event identification. I agree with the points raised by the other reviewers: I appreciate the clarity and simplicity of the approach. All steps are straightforward and easy to understand; nonetheless, I also share some concerns (and it would therefore be useful to provide further information regarding):**

**A**: Thank you for your comment. While we understand that the structure of our manuscript may resemble a short communication, it does not meet the criteria defined by NHESS for that format, which is limited to 2–4 journal pages. Based on the constructive feedback received from all four reviewers, we expanded the manuscript and integrated additional analyses to enhance its scientific depth and clarity.

**R: What is the current early warning system used in Sicily for landslide forecasting, and what is the added value compared to this benchmark? (In this regard, if the system is based solely on triggering rainfall, it would be very interesting to understand the added value of using soil moisture as an event discriminator);**

*A*: Currently there is no early warning system in Sicily specific for landslides. There is indeed an alert system for generic "hydro-geological risk" which is generic for floods and landslides and is based on comparing quantitative precipitation forecasts (QPF) with precipitation depth-duration-frequency curves of different rainfall return periods. Hence, it is based on intrinsic characteristics of rainfall, which is analyzed independently from the observed impacts produced by extreme rainfall. Thus, we added the following consideration within the revised "Results" section:

*"Therefore, satellite or reanalysis-based soil moisture datasets represent a valuable resource to support landslide early warning systems, especially in areas lacking dense monitoring networks. This is particularly relevant in contexts where current warning procedures rely solely on rainfall thresholds, as the inclusion of soil moisture, even coarse or delayed, provides useful information on catchment predisposition and can improve the reliability of alerts."*

**R: what are the characteristics of the movements in the area? It is obviously well known that the added value of using volumetric water content as a proxy depends on the characteristics of the event—events in finer-grained soils are more influenced by antecedent rainfall, while with thinner soil layers or higher permeability, the importance of triggering precipitation increases. With that in mind, is it possible to identify a sort of zonation to understand in which areas there might be added value in using reanalysis-based volumetric soil water content as a proxy? If possible, a spatial representation of the information could be very helpful.**

**A**: We agree with the reviewer that the characteristics of landslide movements can strongly influence the added value of using soil moisture information. In principle, it would be meaningful to perform separate analyses based on the type of movement or according to the geographical alert zones used in Sicily's civil protection system. However, as also noted in our response to Referee #1, the type of movement is unfortunately unknown for most of the 144 landslide events included in our dataset. Additionally, splitting the dataset by the 9 official alert zones would result in very small subsets, which would not be sufficient to support a robust statistical evaluation of model performance. For these reasons, we chose

not to apply a spatial or typological stratification in the present analysis. Nonetheless, we acknowledge the potential value of this approach and consider it a promising direction for future work, particularly as more detailed and spatially distributed landslide information becomes available.

**R: while I understand the rationale for using information with up to a 15-day delay in the first part of the analysis, from the introduction of the ANN approaches onward, I find it less useful and would limit all representations to 5 days before the present time; it is likely that in upcoming releases (ERA6 is expected in December 2025 with 14 km resolution) the delay will decrease and not increase.**

**A**: We understand the reviewer's concern regarding the limited operational relevance of testing latencies beyond 5 days, particularly in the context of ANN-based forecasting. However, we believe that exploring a wider latency window (FROM 1 to 15 days) remains meaningful for two main reasons. First, while the current latency of ERA5-Land is about 5 days and future products such as ERA6 may further reduce this delay, temporary disruptions (e.g., due to maintenance or data access issues) cannot be excluded in operational contexts. Second, from a scientific standpoint, our aim was to investigate the sensitivity of model performance to soil moisture availability over time. This allows us to better understand the degradation of performance as information becomes less timely. We agree that, in practical terms, the most relevant scenarios are within the 1–5 days window, and this is reflected in our discussion of operational perspectives. We appreciate the reviewer's suggestion and the following lines have been added in the revised manuscript.

*"Within this framework, we specifically assess the extent to which different publication lags (up to 15 days) affect the performance of ANN-based landslide prediction. Indeed, although the current latency of ERA5-Land soil moisture data is approximately 5 days, this study explores a broader latency range to evaluate the sensitivity of model performance to delayed information. This choice serves a dual purpose: first, to account for potential improvements in future reanalysis products, e.g., ERA6 (https://climate.copernicus.eu/sites/default/files/2022-09/S3_Hans_Hersbach_v1.pdf) as well as temporary disruptions in data availability due to maintenance or access issues; second, to investigate how the timeliness of soil moisture data affects landslide prediction capability. Understanding the extent to which performance decreases as latency increases allows us to better define the operational value of reanalysis data and identify time thresholds within which delayed soil moisture information remains effective for early warning applications."*

**R: although it is well known to most, for completeness I would include a brief description of the ERA5-Land reanalysis (it would also be helpful to emphasize that 9 km is the resolution of the land module only, while the atmospheric part is a statistical interpolation from the parent model "ERA5"). For accuracy, I would avoid referring to ERA5-Land as "products" or "dataset" and instead use the term "reanalysis outputs."**

*A:* We thank the reviewer for the helpful suggestion. We included a brief description of the ERA5-Land reanalysis to *improve completeness and clarity and to emphasize its relationship with ERA5, as follows:*

*"[...] On the other side, ERA5-Land reanalysis dataset provides soil moisture values ($\vartheta$ [$m^3\ m^{-3}$]) at four distinct depths (i.e., 0–7 cm, 7–28 cm, 28–100 cm, and 100–289 cm), at the hourly scale and at a high spatial resolution. In general terms, ERA5-Land is designed to improve the representation of land surface processes. Its land surface module is natively produced at ~9 km resolution, whereas the atmospheric forcing fields (e.g., precipitation, temperature) are statistically downscaled from the coarser ERA5 reanalysis (31 km) (Hersbach et al., 2020). This structure ensures consistency with ERA5 while enhancing the detail of land surface variables, making ERA5-Land particularly suitable for hydrological and landslide applications."*

Thanks also for your suggestion about the use of correct terminology. However, we have chosen to use the terms "dataset" in accordance with the official terminology adopted by the Copernicus Climate Data Store (CDS), which distributes ERA5-Land (see https://cds.climate.copernicus.eu/datasets/reanalysis-era5-land?tab=overview). Also "reanalysis product" is consistent with the data provider and broader scientific usage (see https://link.springer.com/article/10.1007/s00382-023-06803-w). Thus, we prefer to maintain this terminology in our manuscript.

**R: Additionally, I note: In the abstract, avoid or explain any acronyms used.**

*A: Done*
**R: there's a typo in Figure 1.**
*A: Fixed*